# Barriers and Enablers for the Integration of Industry 4.0 and Sustainability in Supply Chains of MSMEs

Eduardo Machado [1], Luiz Felipe Scavarda [1], Rodrigo Goyannes Gusmão Caiado [1,2,*] and Antonio Márcio Tavares Thomé [1]

1 Industrial Engineering Department, Pontifical Catholic University of Rio de Janeiro (PUC-Rio), Rio de Janeiro 22541-041, Brazil; emachado@eflix.com.br (E.M.); lf.scavarda@puc-rio.br (L.F.S.); mt@puc-rio.br (A.M.T.T.)
2 Tecgraf Institute, Pontifical Catholic University of Rio de Janeiro (PUC-Rio), Rio de Janeiro 22541-041, Brazil
* Correspondence: rodrigocaiado@tecgraf.puc-rio.br

**Abstract:** The integration of Industry 4.0 (I4.0) and sustainability in supply chains emerged as a relevant topic and, therefore, has attracted the interest of academics and practitioners. Many barriers challenge this integration, and enablers to overcome these barriers need to be understood. Micro, Small & Medium Enterprises (MSMEs) have many difficulties to overcome these barriers and successfully implement this integration. Moreover, solutions for larger enterprises do not necessarily fit MSMEs, which reinforces the need to investigate the topic further. Within this context, the goals of this paper are: (i) to identify the main barriers and enablers to integrate I4.0 and sustainability in supply chains of MSMEs and (ii) to analyze the influence among these barriers and enablers, identifying the most prominent ones. A convergent parallel multimethod approach is adopted, first embracing a scoping review to identify main barriers, enablers, and associated categories. Then, conducting a panel of experts with 25 specialists in two rounds to refine and classify the identified barriers and enablers towards the perspective of MSMEs. Finally, two focus group discussions are added using the fuzzy logic and DEMATEL methods to obtain the inter-relationship of barriers and enablers for MSMEs. Research findings reveal eight barriers, eight enablers, and their respective cause-effect relationship, which are expected to help MSMEs managers and decision-makers better understand and implement the integration between I4.0 and sustainability in their supply chains. Results are discussed in eleven research propositions and four propositions for practitioners and policymakers.

**Keywords:** digital transformation; sustainable operations and supply chain management; triple bottom line; developing countries; scoping review; panel of experts; fuzzy-DEMATEL

## 1. Introduction

The implementation of Industry 4.0 (I4.0) technologies has impacted different industries, embracing their supply chains, providing many gains and benefits along the value chains, presenting different challenges for organizations worldwide [1]. In the meanwhile, the theme of sustainability has grown in importance. It presents itself as a permanent reality for enterprises, where practices based on the original concept of triple bottom line (TBL) of sustainability [2], expand its meaning and directly impact their results, development, transformation, and competitiveness [3–6]. Enterprises that include sustainability in their operations directly incorporate the economic, environmental, and social dimensions in their decision-making processes and strategic, tactical, and operational plans [7–10]. Thus, joining sustainable practices and disruptive technologies can improve value chains, and I4.0 aid the implementation of technologies in society [11,12].

Integrating I4.0 technologies, practices related to sustainability, and operations and supply chain management (OSCM) is a global challenge for all kinds of enterprises [1].

Therefore, there is a need to understand barriers that challenge this integration and enablers that can overcome these barriers in the supply chains of different enterprises [13,14]. Moreover, investigations on Micro, Small & Medium Enterprises (MSMEs) are needed on this topic, as MSMEs require different solutions from large enterprises [15,16], but also due to the lack of studies about MSMEs [17,18]. This need is more explicit in developing countries. Their business environments are predominantly made up of MSMEs, with specific structures and legislation to support initiatives to promote employment and generate innovative activities [19]. Moreover, MSMEs are classified as an economic backbone due to their strong position as a generator of jobs [20], despite the weaknesses of the direct relationships between operational skills and sustainability in MSMEs [21].

In the business environment, information technology influence is increasing. Furthermore, sustainability and I4.0 are current concepts for manufacturing, and, notably, the link between these two concepts has a large space in the literature [18,22]. However, research on I4.0 technologies is still highly concentrated in large companies. In contrast, research focused on MSMEs is rare, although all industrial value chains are largely dependent on the contribution of MSMEs as suppliers. For these authors, research needs to address the distinct requirements and conditions present in MSMEs concerning I4.0 characteristics, such as (i) lower digitalization levels; (ii) owner-centered strategic orientation; (iii) more flexible organizational structures. Furthermore, concerning the implementation of I4.0, little was done in MSMEs compared to large organizations, which points out to the likelihood that the I4.0 revolution would take this entire sector by surprise, especially regarding supply chain optimization [17].

There is an even more relevant lack of integration between I4.0 and sustainability [23] in MSMEs. MSMEs are considered one of the most important economic segments worldwide, more specifically, the characteristics and particularities of MSMEs make them special for the economy in an increasingly challenging world [24]. Moreover, it is possible to associate the manufacturing paradigms offered by the I4.0 concept with the challenge of increasing productivity and improving MSMEs performance, and to indicate that it is difficult to deny that MSMEs are frightened by their challenges and have more difficulty overcoming them [24]. A potential solution for MSMEs can be networks and alliances, which help smaller companies overcome resource constraints and improve their ability to take advantage of these new opportunities, brought by the I4.0 potential for a change in value creation [15]. These changes would affect the entire supply chain, from large conglomerates to small suppliers. Furthermore, MSMEs doubt or even fear the concept of I4.0. The results of empirical studies show that MSMEs perceive contextual risks differently, especially because they do not fully adhere to the opportunities of new business models, in contrast to larger enterprises [15]. However, the integration of MSMEs into sustainable supply chains is a key success factor for the implementation of I4.0 technologies and concepts due to the high complexity, speed of development and unpredictability of I4.0. Integrated MSMEs can overcome barriers and adopt enablers inherent in I4.0, with immediate changes in value generation [15]. The central role of senior management in direct support of technological implementation, where managers must align with defined strategies to ensure continuous improvement and sustainability of implementations collaboratively, aiming to facilitate research and development, performance evaluation, and impacts on the supply chain [25]. There is difficulty of MSMEs in accessing I4.0 tools and raise the importance of identifying barriers to implementing I4.0 in MSMEs and the causal relationships between these barriers and the allocation of their scarce resources. Large organizations have scale advantages but counter that I4.0 gives MSMEs operations a special boost that can be better managed with the new age of I4.0 technologies, like industrial internet of things (IIoT), cloud-based manufacturing technology, and big data analytics [16]. There is a need for organizational leaders postural and behavioral change to support the implementation of I4.0. In MSMEs, knowledge sharing has a significant positive impact on the capacity for technological innovation, the introduction of innovation can improve companies' performance, and also, it is necessary to have the financial capacity to introduce

capital-intensive technological innovation [26]. Therefore, the following research questions are posed:

RQ1: What are the main barriers and enablers to integrate I4.0 and sustainability in supply chains from a MSME perspective?

RQ2: What are the causalities and dependencies between these barriers and enablers and the hierarchical levels among them?

Within this context, the goals of this paper are: (i) to identify the main barriers and enablers for the integration of I4.0 and sustainability in supply chains of MSMEs and (ii) to analyze the influence among these barriers and enablers, identifying the most prominent ones. Therefore, the paper aims to provide valuable recommendations to integrate I4.0 and sustainability in supply chains within MSMEs. To this end, a multimethod approach is adopted by combining a scoping review to identify general barriers, enablers, and associated categories available in the literature, a panel of experts with two rounds to refine and classify the identified barriers and enablers towards the perspective of MSMEs, and finally two focus groups using the fuzzy logic and DEMATEL methods to obtain the inter-relationship of the main barriers and enablers for MSMEs. As a result, this study makes manifold contributions to the literature on this topic by identifying and analyzing these barriers and enablers in the context of micro and small as well as medium-sized enterprises, an understanding absent in the current literature and necessary to the smooth integration of I4.0 and sustainability in supply chains. Furthermore, this analysis, based on the Fuzzy-DEMATEL approach, provides useful insights and applications for MSMEs. Adopting the Fuzzy-DEMATEL method is due to its effectiveness in defining the influences among factors (barriers and enablers), in a structural and visual way of cause-and-effect groups, overcoming the inaccuracies inherent in experts' assessments.

This paper is organized into five sections, being the first one the introduction. Section 2 describes the material and methods applied in the research. Section 3 presents the main results of the research, while Section 4 opens the discussions. Finally, Section 5 offers the authors' conclusions and suggestions for future research.

## 2. Materials and Methods

This research adopts a convergent parallel multimethod approach [27] based on three steps: (i) scoping review, (ii) panel of experts, and (iii) focus groups, as shown in Figure 1. In convergent parallel multimethod designs, the researcher combines almost simultaneously qualitative and quantitative methods, triangulating methods and data in the search of a deeper understanding of the phenomenon under study. The broad categories for the identification of enablers and barriers were identified in step 1. Sequentially, a panel of experts was convened to refine the typology and to identify antecedents and consequents to rank barriers and enablers identified in phase 1. The focus group took place sequentially after step 2 to clarify issues of consensus and disagreements among judges. Each step is detailed throughout this Section 2. The rationale to transition from one step to the next is explained for each step below.

### 2.1. Scoping Review

Scoping reviews can quickly map the key concepts that underpin an area of research and the main sources and evidence available in terms of its nature, characteristics, and volume, focusing on broader items with multiple study applications [28,29]. Scoping reviews are particularly applicable when the literature has not yet been comprehensively reviewed or exhibits a large, complex, or heterogeneous nature, as the one targeted in this research. According to [30], scoping is more general than systematic reviews by design: "a key difference between scoping reviews and systematic reviews is that in terms of a review question, a scoping review will have a broader "scope" than traditional systematic reviews with correspondingly more expansive inclusion criteria." Therefore, this research step aims to scan the literature available and then picks out the barriers and enablers from work already carried out, as in [31]. Additionally, research gaps are identified. Thus, the gaps

and, consequently, the research questions (derived from the gaps pointed out in Section 1) that guided this study, are identified and sorted according to a scoping review [28]. The purpose of the scoping review is to identify and analyze knowledge gaps, to identify the types of available evidence in the field of I4.0 and SSCM, and to identify barriers and enablers (factors) [30], related to the concept of the integration of I4.0 and sustainability in supply chains of MSMEs.

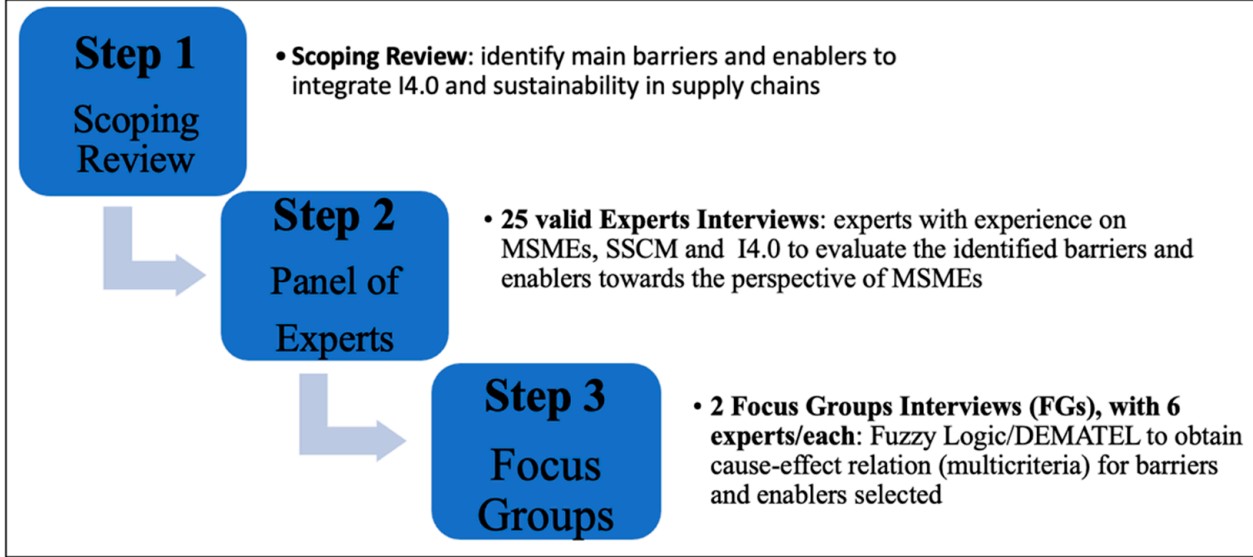

**Figure 1.** Research steps.

This research conducts as its first step an exploratory scoping review following [28] guideline, consisting of the following steps: (i) Research questions definition; (ii) Relevant studies identification; (iii) Studies' selection; (iv) Data Mapping; (v) Grouping and summarization of results.

The introduction section of the paper offers the problem definition and the research scope, questions, and goals, addressing step 1. Following [32,33], two databases were chosen to retrieve relevant studies. Web of Science and Scopus were chosen as they cover similar research domains and are complementary [34], besides being relevant for sustainability and OM topics [3,35]. The following combination of search keywords was applied to the title, abstract and keywords of papers available in these two databases to identify studies: [("Industry 4.0" or "Smart manufacturing") AND ("sustainab*" or "green") AND ("supply chain" or "SCM")]. The Preferred Reporting Items tool for Systematic Reviews and Meta-Analyses (PRISMA) statement is used, based on the studies by [36] to help systematic reviewers to report the process transparently. It includes concepts of [37] with new reporting guidelines that reflect advances in methods for identifying, selecting, evaluating, and synthesizing studies. Figure 2 presents the results. The research identified initially 202 studies, which dropped to 169 after removing duplicates from the two databases. The study selection counted exclusively with studies written in the English language and published up to September 2020. The snowball approach was applied following [32] in the period up to January 2021, both through backward and forward searches. The studies' selection excludes studies not published in peer-reviewed journals and not related to enablers and barriers to integrating I4.0 and sustainability in OSCM. In the end, 27 studies were selected and mapped for further analysis.

Results were grouped and summarized, aided by a narrative synthesis and content analysis [32,38], developing categories for main barriers and enablers from a detailed examination of all selected studies.

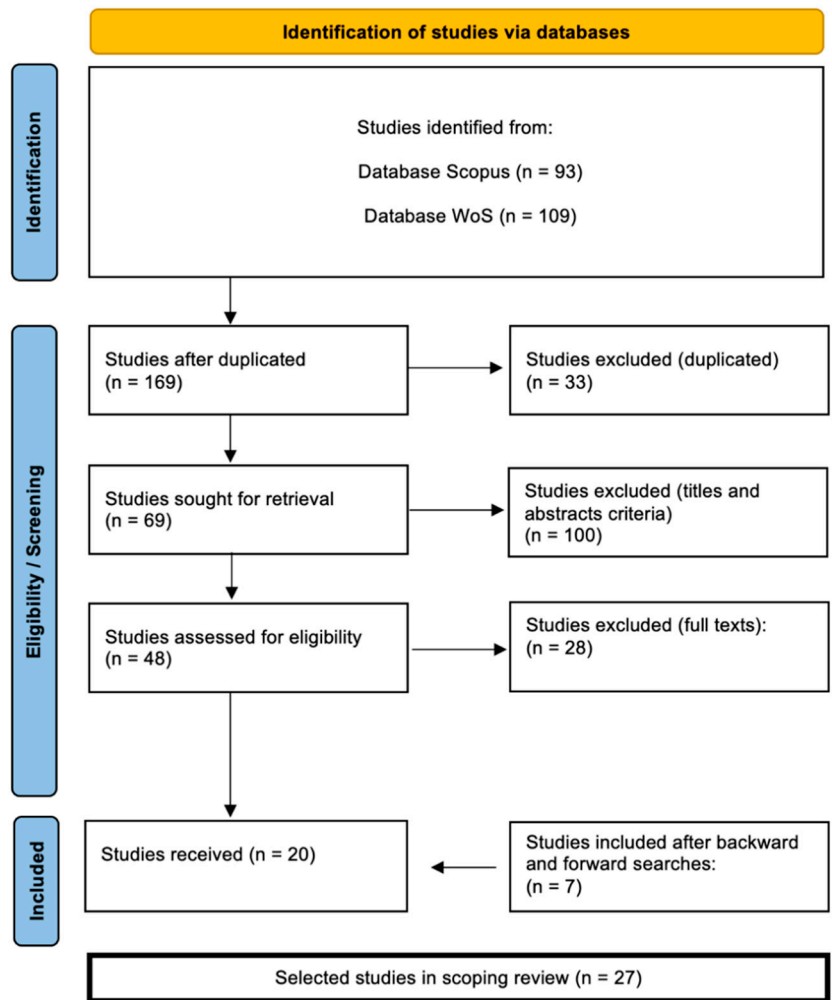

**Figure 2.** Studies retrievals from the PRISMA perspective: Adapted from ref. [37].

### 2.2. Panel of Experts

In this methodological step, the revealed list of barriers and enablers obtained from the scoping review and their respective categories were used as the basis for the first round of individual interviews through a questionnaire. On the expert panel, anonymous responses are distributed to experts, who are allowed to review their own responses in subsequent rounds until consensus is reached [39]. Based on the perception of specialists in MSMEs, this step seeks to filter and refine barriers and facilitators (factors), where the main ones will be identified [40]. This step is supported by experts to validate the list of barriers and facilitators (factors) and to find other factors based on professional views and experiences [31]. Thus, the identified factors are complemented through expert panels [40], through a specific questionnaire aimed at grouping and selecting factors that adhere to the reality of MSMEs in a developing country.

Appendix A presents the questionnaire used. Section 1 includes the informed consent term, objectives, procedures, and a glossary. The second section regards the expert profile, while the third section regards the barriers and the fourth the enablers. Initially, barriers and enablers are listed from the application of the primary questionnaire through the first round of individual interviews. Importantly, this questionnaire contains open questions for barriers and enablers. Experts were allowed to indicate new barriers and enablers that they perceived were missing in the initial relationship and were relevant for MSMEs. These new barriers and enablers were the bases for the second round with the experts. Appendix B builds upon the questionnaire of Appendix A, now customized for the new evidence brought from the experts.

Thirty individual interviews with experts were carried out, five of which were disregarded as the evidence obtained with experts with no experience in I4.0 or SSCM or MSMEs were not considered. Therefore, evidence from 25 specialists were considered for future analysis. Experts from Brazil were chosen for the research, associating this country as a good representative for developing countries [10,41]. Each specialist indicated the predominance of own experience in MSMEs in the options 'micro and small companies', 'medium companies', or 'All (micro and small, and medium companies)'. As MSMEs generally have a particular and focused business model, allowing generalizable statements for the entire enterprise [18,42,43], a single respondent was involved in each company. This is difficult to occur in larger organizations, as statements are limited to the perspective of an informant who cannot capture fully organizational aspects. However, MSMEs are especially suited to be investigated with interviews that cover a single informant per company [18,42,43], as conducted in the panel of experts for this research.

A Likert scale questionnaire was adopted. The questionnaire followed the set of five guidelines for the construction of Likert scale instruments from [44]: (i) understanding of the construct; (ii) item development; (iii) determination of the results space; (iv) specification of the classification model; (v) collection of feedback and pilot testing of the questionnaire. The objective in following these guidelines was to develop questionnaires using the Likert scale, in this case, a seven items scale, to produce compliant data and, consequently, which valid interpretations, making it possible to indicate both barriers and enablers ranked in the following options: (_) None (_) Very Low (_) Low (_) Medium (_) High (_) Very High (_) Fully. The panel of experts was applied with follow-up and individual guidance of one of the authors of this paper. Before answering the questionnaire, each specialist had a meeting with at least one author to clarify scope, procedures and resolve doubts. The average duration of the first round of the panel of experts was 1h15 (Appendix A) and the second round lasted 15min (Appendix B). All specialists joined both rounds.

### 2.3. Focus Group

With focus groups, this study seeks consensus, considering focus groups as fundamental units of analysis through homogeneous participants and a moderator prepared to stimulate constructive discussion [45]. Focus groups are appropriate, as they allow questions aimed at assessing both the influence and priority of barriers and facilitators in the context of MSMEs. They also provide an environment for in-depth discussion to help formulate research proposals. Focus groups are similar to panel of experts, however, they are structured for verbal responses and exchanges rather than in writing. Thus, everyone in the group is aware of the origin of the answers. The group is given a set of questions, usually before the meeting. The facilitator asks the questions and allows each member to express their opinion. Discussion is allowed, stimulated, and controlled by the facilitator, always with the objective of obtaining consensus [39]. Focus groups capture the participants' experiences, observations, and opinions, allowing researchers to perceive value during and after conducting the activities [45]. Therefore, new information that was not available before provides a greater understanding of the topics covered, refining barriers and enablers from the MSMEs perspective. Focus groups also allow researchers to address findings more accurately, enabling a better understanding of the phenomena. Consensus is the main characteristic produced in work with focus groups, overriding the individual views of the participants [46]. Therefore, this method was adopted for the third step of this research.

Two focus groups were formed with six experts in each, with experience in the three items covered: (i) SSCM; (ii) I4.0; (iii) MSMEs. In these three items, weight of 1/3 (one third or 33.33%) of the total weight 1 (one or 100%) was applied, adopting a three-level criterion, following the parameters presented in Table 1.

**Table 1.** Experts Experience and Weights.

| Experience | | | Weights | |
|---|---|---|---|---|
| SSCM | | | 33.33% | |
| I4.0 | | | 33.33% | |
| MPMEs | | | 33.33% | |
| SSCM | | I4.0 | | MPMEs |
| Level 3 (10+ years) | 3 | Level 3 (6+ years) | 3 | Level 3 (21+ years) | 3 |
| Level 2 (5–9 years) | 2 | Level 2 (3–5 years) | 2 | Level 2 (11–20 years) | 2 |
| Level 1 (up to 4 years) | 1 | Level 1 (up to 2 years) | 1 | Level 1 (up to 10 years) | 1 |

To quantify the relationship between evaluation factors, they give their linguistic assessments in the forms of intuitionistic fuzzy sets representing which factors have direct relation with each other [47]. The DEMATEL method identifies the cause-effect relationship of the criteria related to the problem, being effective to visualize the structure of complicated causal relationships with matrices or digraphs, which in turn portray a contextual relationship between the elements of the system, in which a numeral represents the force of influence, making the relationship structured and intelligible. To establish a structural model of experts' judgments, their respective preferences and importance are attributed with notable values, which in turn are inadequate in the real world as they are often obscure and difficult to estimate by exact numerical values, thus creating the need of fuzzy logic [48].

The cause-effect relationship among the barriers and enablers was revealed aided by the DEMATEL method. It is a recognized and widely used method to adequately represent the causal relationship between the complex elements of a system. Each of the experts involved in the focus group uses a linguistic term converted in a number from 0 (zero) to 4 (four) that has a corresponding linguistic value (triangular fuzzy number), as presented on Table 2, based on [49].

**Table 2.** The correspondence of linguistic terms and values.

| Linguistic Term | Crisp Value | Fuzzy Number |
|---|---|---|
| Very high influence (VH) | 4 | (0.75, 1.0, 1.00) |
| High influence (H) | 3 | (0.50, 0.75, 1.00) |
| Low influence (L) | 2 | (0.25, 0.50, 0.75) |
| Very low influence (VL) | 1 | (0.00, 0.25, 0.50) |
| No influence (N) | 0 | (0.00, 0.00, 0.25) |

Two matrices were created from the assessments carried out by experts, with the main barriers and enablers of each of the identified categories. Each of the matrices addressed a specific enterprise group: (i) micro and small enterprises (MSEs); (ii) medium enterprises (MEs). Each category was represented by their main barrier or facilitator. The six experts from each focus group reached a consensus so that the assessment could be entered in the spreadsheet. The first focus group of specialists with a concentration of accumulated experience in micro and small-sized companies, lasted 03 h and 15 min and a consensus obtained in most cases without the need for intervention by the interviewer, who consisted of one of the authors of this paper, acting as a mediator. The second focus group of specialists with a concentration of accumulated experience in medium-sized companies, lasted 2 h and 30 min, and a consensus was also obtained in most cases without the need for intervention by the interviewer.

In this research, fuzzy set theory has also been embedded with DEMATEL to overcome the inaccuracy and bias of experts' decisions [50]. The Fuzzy-DEMATEL method was applied to assess the causal relationship between barriers and enablers using expert input.

According to [51], expert assessments contain inaccuracies and subjectivity, and the fuzzy theory directly addresses this issue as indicated by [52]. Based on [49] and [53], DEMATEL has an advantage over other methods of multicriteria decision-making (MCDM) and interpretive structuring modeling (ISM), as it reveals the interrelationships between factors, prioritizes and separates them into cause-and-effect groups. The presentation of results in the form of matrices and graphs is also efficient in dealing with the complex cause-effect relationships between the factors, which in the case of this research, are the barriers and enablers. Therefore, the Fuzzy-DEMATEL was applied in this research based on the adopted steps offered in [49,53–56] as follows:

(i)     Create the correspondence of linguistic terms and values (see Table 2);
(ii)    Aggregate results and obtain a fuzzy pairwise direct-relation matrix (X);
(iii)   Normalize the direct-relation matrix and calculate the total relation matrix (T);
(iv)    Determine row and column sums from T;
(v)     Determine the overall prominence and net effect values of factors (D + R and D − R);
(vi)    Formulate the DEMATEL cause-effect diagrams. Each step incorporates multiple mathematical evaluations. The prominence and net effect values of each factor are fuzzy-DEMATEL analysis outputs [55].

The final prominence value ranks the factors. Additional details on the fuzzy-DEMATEL methodology and the calculations appear in Table 2 and Appendix C.

From an 'n × n' identity matrix T, we have R being the sum of the rows and D being the sum of the columns of the matrix T, D + R is set to highlight, indicating the prominence of factors in the system, and D − R represents the importance for the influence of each factor (Figure 3).

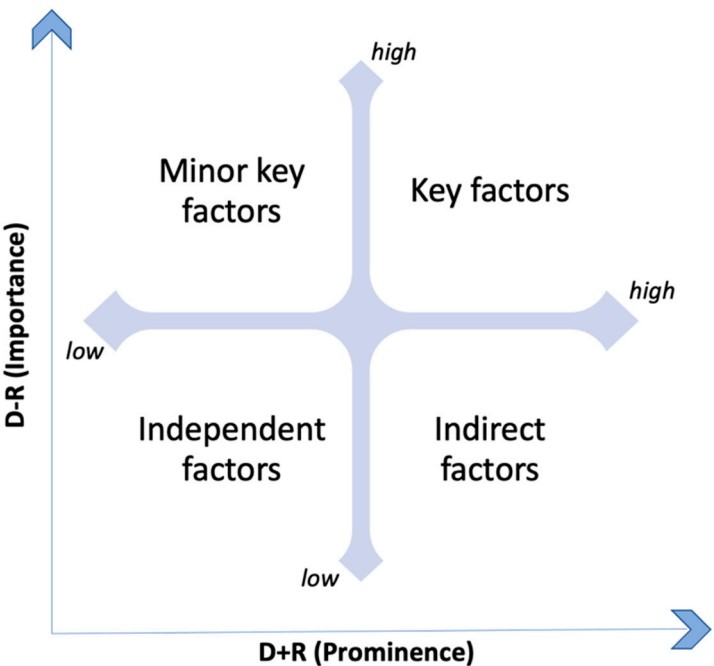

**Figure 3.** The DEMATEL MAP: Adapted from ref. [47].

## 3. Results

Table 3 presents the 13 barriers (B) and 31 enablers (E) retrieved from the scoping review, added by the three extra barriers (EB) and two extra enablers (EE) obtained with the panel of experts. Appendix D presents the evaluation of the experts within the panel for each barrier and enabler. During this process, there was no purpose to balance the number of barriers and of enablers, or to develop a symmetry, but to retrieve the main ones available in the literature and refined with the aid of experts.

**Table 3.** Main barriers and enablers to sustainable I4.0 in MSMEs.

| Barrier (B) or Barrier-Extra (BE) (Total = 16) | Enabler (E) or Enabler-Extra (EE) (Total = 33) |
|---|---|
| B1—Cybersecurity issues/B2—Cost of improvement & OSCM economic condition/B3—Lack of support from regulatory authority/poor legislation/B4—Lack of commitment from top management/B5—Interoperability issues/B6—Employability/B7—Lack of technical expertise/B8—Alternative resources and energy needs/B9—Design challenges to reuse and recovery products/B10—Organizational barriers, Capacity constraints/B11—Resistance to change/change management practices and adopting innovation/for society/B12—Data management and storage issues/B13—Inequalities of opportunities and digitalization risks/BE14—Cultural aspects/BE15—Sub-utilization of academic knowledge/universities/BE16—Lack of investment in R&D | E1—Internal innovation process/E2—Open innovation/E3—Change management/E4—Dynamic capabilities/E5—Innovative business models and service design/E6—Customer and supplier integration/E7—Support of unconventional partners/E8—Governmental and institutional pressures/E9—Collaborative networks/E10 -Innovative business models and service design techniques/E11—Re-designing and decentralized structure/E12—Strategic alignment/E13—I4.0 readiness/E14—Adoption of smart factory solutions/E15—Data-centered solutions/E16—Consistent data flow/E17—Modular design/E18—Information transparency and data security/E19—Sustainable philosophy/E20—Focus on renewable natural resources/E21—Interdisciplinary and holistic integration/E22—Sharing economy/E23—Life cycle thinking and circular processes/E24—Knowledge sharing/E25—Effective communication/E26—Individual incentive schemes/E27—Employee's empowerment/E28—Experimentation/E29—Education and training focused on soft and technical skills/E30—Transformational leadership/E31—Top management commitment/EE32—Better management of Certifications, Standards and Regulations/EE33—Valuing R&D and Research Centers |

A complete analysis of all barriers and enablers for categorization was carried out through content analysis resulting in eight symmetrical categories: people, technology, innovation, institutional, OSCM related topics, legal, organization and sustainability.

Concerning the barriers, eight out of the 16 barriers retrieved were selected. In contrast, for the enablers, eight out of the 33 enablers were selected, with a grouping of five more (by similarities) enablers in categories: organization (1+), people (2+), technology (1+) and sustainability (2+), respectively, according to Tables 4 and 5. It is noteworthy that the hierarchical criteria and the grouping into categories based on the assessments carried out by experts are respected.

**Table 4.** Barriers selection for 8 × 8 matrices.

| Classification | Selected | Categories |
|---|---|---|
| B1 | Lack of technical expertise | People |
| B2 | Cybersecurity issues | Technology |
| B3 | Resistance to change/change management practices and adopting innovation for society | Innovation |
| B4 | Lack of investment in R&D | Institutional |
| B5 | Cost of improvement & OSCM economic condition | OSCM related topics |
| B6 | Lack of support from regulatory authority/poor legislation | Legal |
| B7 | Lack of commitment from top management | Organization |
| B8 | Alternative resources and energy needs | Sustainability |

Once the matrices were stablished for both selected barriers and enablers, paired comparisons were carried out in two focus groups with specialists divided according to the size of the MSMEs. Figures 4 and 5, respectively, for barriers and enablers for MSEs, report the classification made by the specialists who declared a predominance of accumulated experience in MSEs. Figures 6 and 7, respectively, for barriers and enablers for MEs, report the classification made by the specialists who declared a predominance of accumulated

experience in MEs. Figures 4–7 depict the internal influence among barriers (similarly among enablers), separately for MSEs and MEs. Reading rows in relation to columns denotes how much does say B1 influence B2. For focus group 1 in Figure 4 (MSEs) it would be a very high influence (VH). Reading columns in relation to lines inform influencers of a given barrier, say how much is B2 influenced by B5. For focus group 1 in Figure 4 (MSEs) it would be a very high influence (VH), and for focus group 2 (MEs) in Figure 6 it would be a high influence (H). The verbal scales are depicted graphically for the main influencers in Section 4 (Discussion).

**Table 5.** Enablers selection for 8 × 8 matrices.

| Classification | Selected | Categories |
|---|---|---|
| E1 | Top management commitment + Strategic alignment | Organization |
| E2 | Employees' empowerment + Knowledge sharing + Effective communication | People |
| E3 | Internal innovation process | Innovation |
| E4 | Data-centered solutions + Consistent data flow | Technology |
| E5 | Interdisciplinary and holistic integration + Life cycle thinking and circular processes | Sustainability |
| E6 | Customer and supplier integration | OSCM related topics |
| E7 | Governmental and institutional pressures | Legal |
| E8 | Valuing R&D/Research Centers | Institutional |

| Barrries (Focus Group Micro & Small Enterprises) | B1 | B2 | B3 | B4 | B5 | B6 | B7 | B8 |
|---|---|---|---|---|---|---|---|---|
| B1 | ■ | VH | VH | VH | VH | VL | L | H |
| B2 | L | ■ | VL | VL | H | L | L | H |
| B3 | H | H | ■ | VH | H | L | H | L |
| B4 | H | H | L | ■ | VH | VH | L | L |
| B5 | VH | VH | VH | VH | ■ | VL | H | L |
| B6 | H | VH | H | L | L | ■ | VL | VL |
| B7 | VH | VH | VH | VH | H | VL | ■ | L |
| B8 | VL | H | H | H | VH | L | L | ■ |

**Figure 4.** The correspondence of linguistic terms and values—MSEs (Barriers).

| Enablers (Focus Group Micro & Small Enterprises) | E1 | E2 | E3 | E4 | E5 | E6 | E7 | E8 |
|---|---|---|---|---|---|---|---|---|
| E1 | ■ | VH | VH | VH | VH | H | L | H |
| E2 | VH | ■ | VH | VH | H | H | H | H |
| E3 | L | VH | ■ | VH | VH | H | L | H |
| E4 | VH | H | VH | ■ | H | VH | L | VH |
| E5 | H | H | H | H | ■ | H | H | H |
| E6 | H | L | H | VH | VH | ■ | VH | H |
| E7 | H | L | L | H | VH | VH | ■ | VH |
| E8 | L | VH | VH | VH | H | H | L | ■ |

**Figure 5.** The correspondence of linguistic terms and values—MSEs (Enablers).

| Barriers (Focus Group Medium Enterprises) | B1 | B2 | B3 | B4 | B5 | B6 | B7 | B8 |
|---|---|---|---|---|---|---|---|---|
| B1 | | VH | VH | VH | VH | VL | H | VH |
| B2 | VL | | H | L | H | H | VL | VH |
| B3 | H | VH | | VH | H | VL | VH | H |
| B4 | VH | L | H | | H | VL | VL | H |
| B5 | VH | H | H | H | | VL | H | H |
| B6 | VL | H | VL | H | H | | H | VL |
| B7 | VH | VH | VH | VH | VH | VL | | H |
| B8 | H | VH | H | VH | H | VL | H | |

**Figure 6.** The correspondence of linguistic terms and values—MEs (Barriers).

| Enablers (Focus Group Medium Enterprises) | E1 | E2 | E3 | E4 | E5 | E6 | E7 | E8 |
|---|---|---|---|---|---|---|---|---|
| E1 | | VH | VH | VH | VH | VH | VH | VH |
| E2 | L | | VH | VH | VH | L | VH | VH |
| E3 | L | VH | | H | VH | H | L | VH |
| E4 | H | H | VH | | L | H | VL | VH |
| E5 | H | H | L | L | | VH | VH | H |
| E6 | VH | H | H | L | H | | H | VH |
| E7 | VH | VH | H | H | H | H | | H |
| E8 | VH | VH | VH | VH | H | VH | L | |

**Figure 7.** The correspondence of linguistic terms and values—MEs (Enablers).

Table 6 and Figure 8 present the Fuzzy-DEMATEL Approach, Tables and Diagrams results of cause-effect relationships for barriers in MSEs. Barriers B1, B6, B7 and B8 are in the Cause Group, while barriers B2, B3, B4 and B5 are in the Effect Group.

**Table 6.** $D + R/D - R$ Results for Barriers (MSEs).

| Barrier | Di + Ri | Di − Ri | Cause(C)/Effect(E) |
|---|---|---|---|
| B1—Lack of technical expertise | 9.490042 | 0.079289 | C |
| B2—Cybersecurity issues | 9.04908 | −1.515973 | E |
| B3—Resistance to change/change management practices and adopting innovation for society | 9.392564 | −0.040761 | E |
| B4—Lack of investment in R&D | 9.430682 | −0.271205 | E |
| B5—Cost of improvement & OSCM economic condition | 9.956615 | −0.38204 | E |
| B6—Lack of support from regulatory authority/poor legislation | 7.432845 | 0.367826 | C |
| B7—Lack of commitment from top management | 8.840327 | 0.698012 | C |
| B8—Alternative resources and energy needs | 8.327192 | 0.245782 | C |

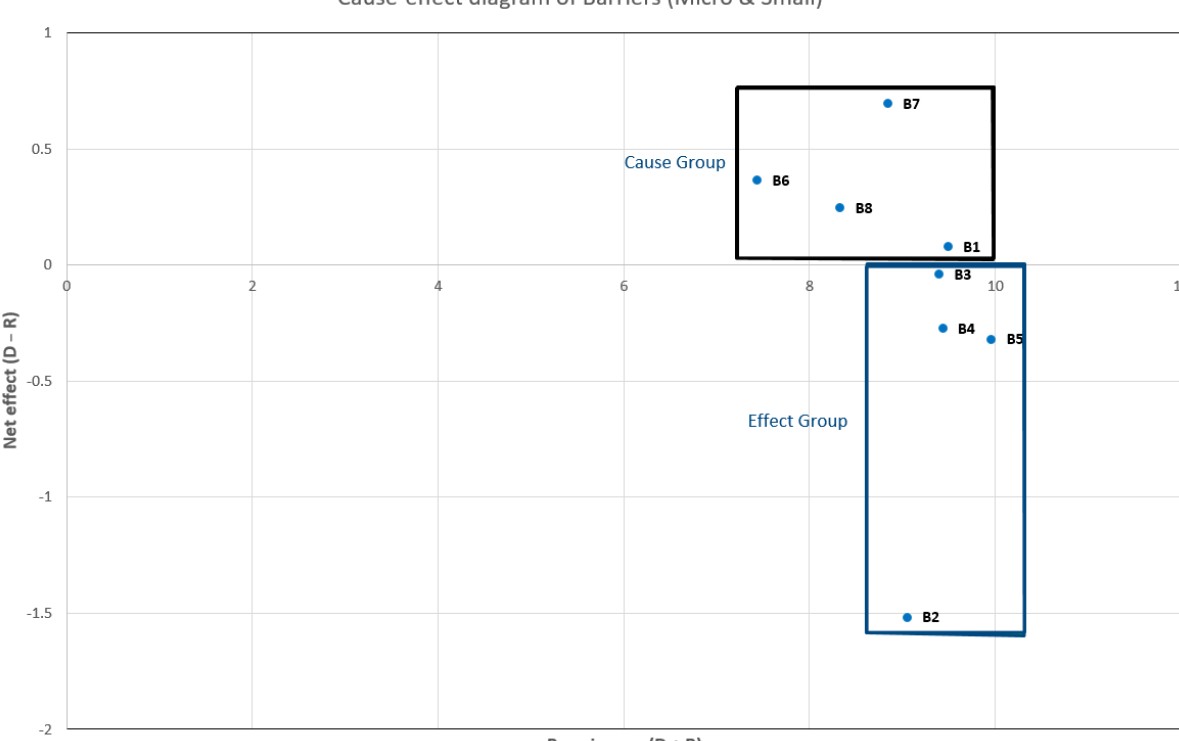

**Figure 8.** Cause-Effect Diagram of Barriers (MSEs).

Table 7 and Figure 9 present cause-effect relationships for enablers in MSEs. Enablers E1, E2 and E7 are in the Cause Group, while enablers E3, E4, E5, E6 and E8 are in the Effect Group.

**Table 7.** D + R/D − R Results for Enablers (MSEs).

| Enabler | Di + Ri | Di − Ri | Cause(C)/Effect(E) |
|---|---|---|---|
| E1—Top management commitment + Strategic alignment | 15.04984 | 0.43318 | C |
| E2—Employee's empowerment + Knowledge sharing + Effective communication | 15.3215 | 0.41254 | C |
| E3—Internal innovation process | 15.21509 | −0.359365 | E |
| E4—Data-centered solutions + Consistent data flow | 15.92757 | −0.322837 | E |
| E5—Interdisciplinary and holistic integration + Life cycle thinking and circular processes | 15.48851 | −0.415236 | E |
| E6—Customer and supplier integration | 15.37755 | −0.124541 | E |
| E7—Governmental and institutional pressures | 13.99828 | 0.657049 | C |
| E8—Valuing R&D/Research Centers | 15.15782 | −0.322529 | E |

Table 8 and Figure 10 present the cause-effect relationships for barriers in MEs. Barriers B1, B6 and B7 are in the Cause Group, while barriers B2, B3, B4, B5 and B8 are in the Effect Group.

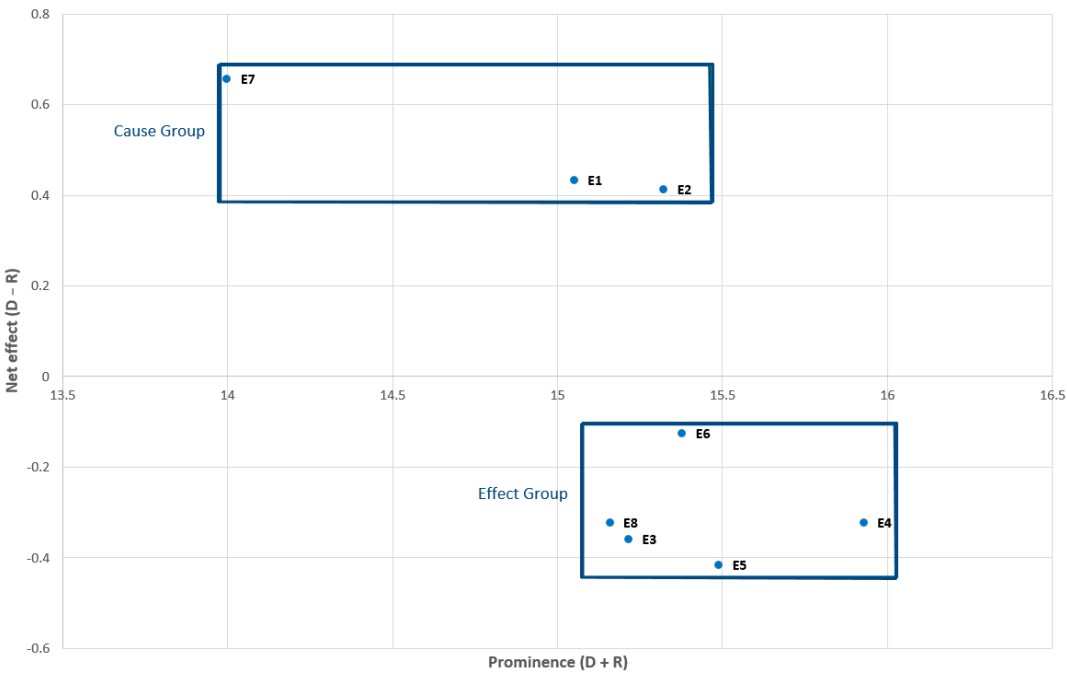

**Figure 9.** Cause-Effect Diagram of Enablers (MSEs).

**Table 8.** D + R/D − R Results for Barriers (MEs).

| Barrier | Di + Ri | Di − Ri | Cause(C)/Effect(E) |
|---|---|---|---|
| B1—Lack of technical expertise | 13.7021 | 0.4476 | C |
| B2—Cybersecurity issues | 13.068 | −1.2377 | E |
| B3—Resistance to change/change management practices and adopting innovation for society | 13.8261 | −0.0799 | E |
| B4—Lack of investment in R&D | 13.1995 | −1.1248 | E |
| B5—Cost of improvement & OSCM economic condition | 13.9027 | −0.5104 | E |
| B6—Lack of support from regulatory authority/poor legislation | 9.40544 | 1.28372 | C |
| B7—Lack of commitment from top management | 13.2573 | 0.84586 | C |
| B8—Alternative resources and energy needs | 13.7163 | −0.2106 | E |

Table 9 and Figure 11 present the cause-effect relationships for enablers in MEs. Enablers E1 and E7 are in the Cause Group, while enablers E2, E3, E4, E5, E6 and E8 are in the Effect Group.

**Table 9.** D + R/D − R Results for Enablers (MEs).

| Enabler | Di + Ri | Di − Ri | Cause(C)/Effect(E) |
|---|---|---|---|
| E1—Top management commitment + Strategic alignment | 14.5263 | 0.89105 | C |
| E2—Employee's empowerment + Knowledge sharing + Effective communication | 14.3895 | −0.3045 | E |
| E3—Internal innovation process | 13.9268 | −0.329 | E |

**Table 9.** *Cont.*

| Enabler | Di + Ri | Di − Ri | Cause(C)/Effect(E) |
|---|---|---|---|
| E4—Data-centered solutions + Consistent data flow | 13.2389 | −0.3082 | E |
| E5—Interdisciplinary and holistic integration + Life cycle thinking and circular processes | 13.71 | −0.2877 | E |
| E6—Customer and supplier integration | 13.9409 | −0.0458 | E |
| E7—Governmental and institutional pressures | 13.4648 | 0.75641 | C |
| E8—Valuing R&D/Research Centers | 14.7837 | −0.1847 | E |

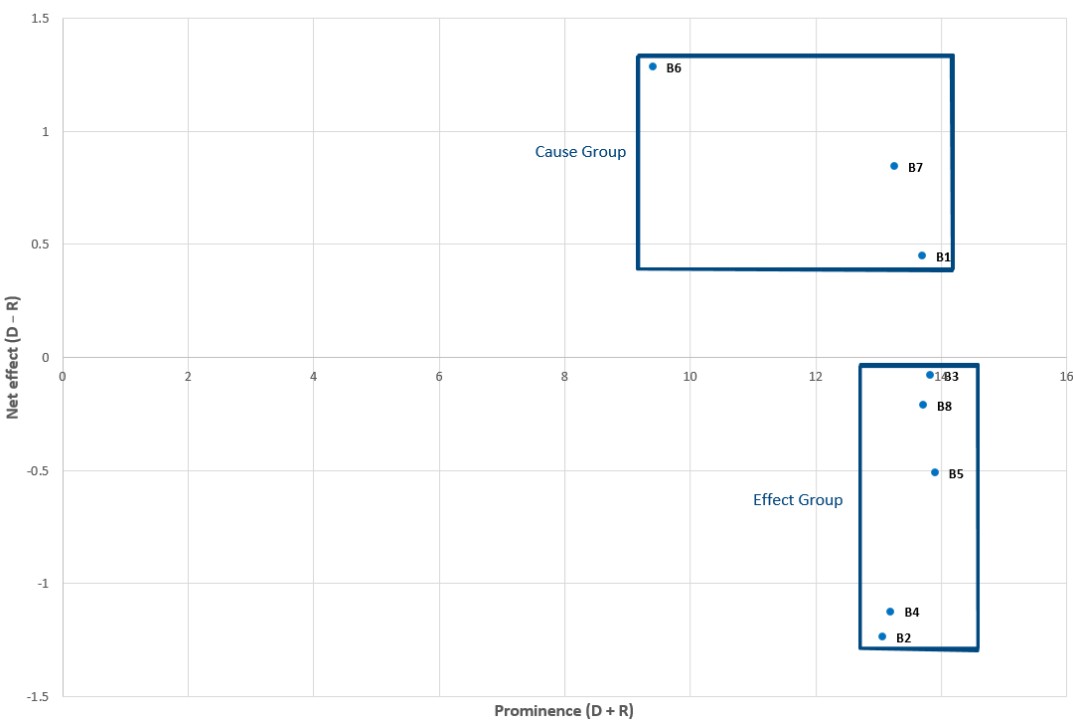

**Figure 10.** Cause-Effect Diagram of Barriers (MEs).

Table 10 and Figure 12 present results of cause-effect relationships for barriers in MSMEs companies (aggregated focus groups). Barriers B1, B6, B7 and B8 are in the Cause Group, while barriers B2, B3, B4 and B5 are in the Effect Group.

**Table 10.** D + R/D − R Results for Barriers (MSMEs)—Aggregated Focus Groups.

| Barrier | Di + Ri | Di − Ri | Cause(C)/Effect(E) |
|---|---|---|---|
| B1—Lack of technical expertise | 11.1200 | 0.2702 | C |
| B2—Cybersecurity issues | 10.6307 | −1.3595 | E |
| B3—Resistance to change/change management practices and adopting innovation for society | 11.1267 | −0.0358 | E |
| B4—Lack of investment in R&D | 10.8968 | −0.6373 | E |
| B5—Cost of improvement & OSCM economic condition | 11.4957 | −0.4074 | E |

**Table 10.** *Cont.*

| Barrier | Di + Ri | Di − Ri | Cause(C)/Effect(E) |
|---|---|---|---|
| B6—Lack of support from regulatory authority/poor legislation | 8.1971 | 0.7427 | C |
| B7—Lack of commitment from top management | 10.6034 | 0.7560 | C |
| B8—Alternative resources and energy needs | 10.4919 | 0.0471 | C |

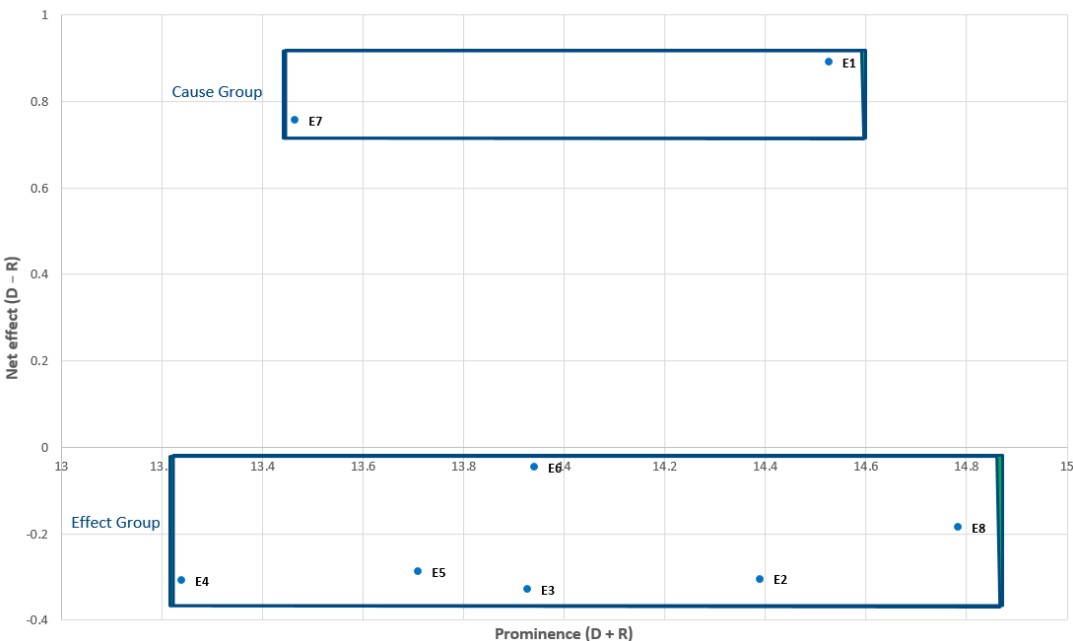

**Figure 11.** Cause-Effect Diagram of Enablers (MEs).

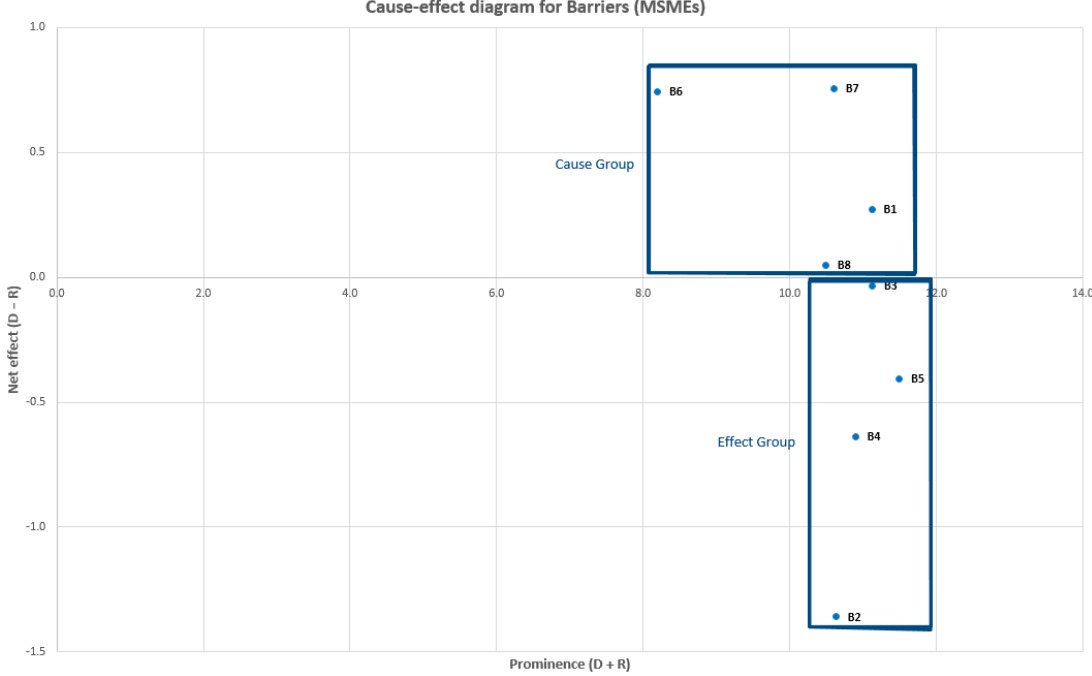

**Figure 12.** Cause-Effect Diagram of Barriers (MSMEs)—Aggregated Focus Groups.

Table 11 and Figure 13 present the cause-effect relationships for enablers in MSMEs companies (aggregated focus groups). Enablers E1, E2 and E7 are in the Cause Group, while enablers E3, E4, E5, E6 and E8 are in the Effect Group.

**Table 11.** D + R/D − R Results for Enablers (MSMEs)—Aggregated Focus Groups.

| Enabler | $D_i + R_i$ | $D_i - R_i$ | Cause(C)/Effect(E) |
|:---|:---:|:---:|:---:|
| E1—Top management commitment + Strategic alignment | 14.61628 | 0.630341 | C |
| E2—Employee's empowerment + Knowledge sharing + Effective communication | 14.68787 | 0.02143 | C |
| E3—Internal innovation process | 14.39422 | −0.365395 | E |
| E4—Data-centered solutions + Consistent data flow | 14.40495 | −0.328144 | E |
| E5—Interdisciplinary and holistic integration + Life cycle thinking and circular processes | 14.40467 | −0.379746 | E |
| E6—Customer and supplier integration | 14.47304 | −0.113201 | E |
| E7—Governmental and institutional pressures | 13.58132 | 0.69578 | C |
| E8—Valuing R&D/Research Centers | 14.79827 | −0.274152 | E |

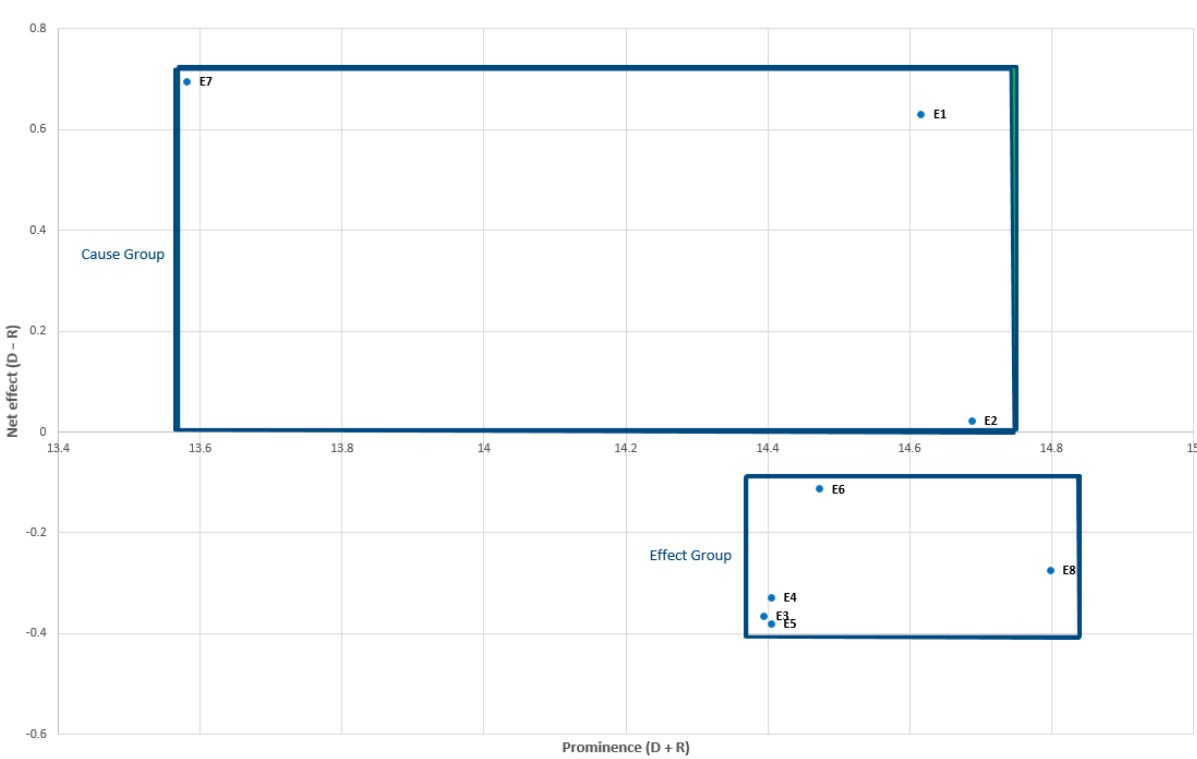

**Figure 13.** Cause-Effect Diagram of Enablers (MSMEs)—Aggregated Focus Groups.

Based on Figure 3 (The DEMATEL map), barriers and enablers are distributed into four classifications: (i) key; (ii) minor key; (iii) indirect; (iv) independent. Table 12 presents the distribution for MSEs, MEs and for MSMEs, which is the central focus for analysis in this topic.

**Table 12.** The DEMATEL map discussion for Barriers and Enablers.

| Barriers | MSEs | MEs | MSMEs (Combined) |
|---|---|---|---|
| Key barriers | B1, B6, B7, B8 | B1, B6, B7 | B1, B6, B7, B8 |
| Minor key barriers | none | none | none |
| Indirect barriers | B2, B3, B4, B5 | B2, B3, B4, B5, B4, B8 | B2, B3, B4, B5 |
| Independent barriers | none | none | none |
| **Enablers** | **MSEs** | **MEs** | **MSMEs (Combined)** |
| Key enablers | E1, E2 | E1 | E1, E2 |
| Minor key enablers | E7 | E7 | E7 |
| Indirect enablers | E3, E4, E5, E6, E8 | E2, E8 | E3, E4, E5, E6, E8 |
| Independent enablers | none | E3, E4, E5, E6 | none |

Note: B1—Lack of technical expertise; B2—Cybersecurity issues; B3—Resistance to change/change management practices and adopting innovation for society; B4—Lack of investment in R&D; B5—Cost of improvement & OSCM economic condition; B6—Lack of support from regulatory authority/poor legislation; B7—Lack of commitment from top management; B8—Alternative resources and energy needs. E1—Top management commitment + Strategic alignment; E2—Employee's empowerment + Knowledge sharing + Effective communication; E3—Internal innovation process; E4—Data-centered solutions + Consistent data flow; E5—Interdisciplinary and holistic integration + Life cycle thinking and circular processes; E6—Customer and supplier integration; E7—Governmental and institutional pressures; E8—Valuing R&D/Research Centers.

The classification of barriers does not show much variation between MSEs, MEs, and combined MSMEs. B8 appears as an indirect barrier for MEs and as a key barrier both for MSEs and in combined MSMEs, where in addition to B8, barriers B1, B6 and B7 also appear as key barriers for MSMEs. B8 is classified as an indirect barrier in MEs. It is noteworthy that there is no classification of barriers as minor key barriers or independent barriers.

Regarding enablers, there is full alignment between the results for MSEs and combined MSMEs. E1 is a key enabler. E2 appears as key enabler for MSEs and combined MSMEs, and as indirect enabler for MEs. E3, E4, E5 and E6 appear as independent enablers for MEs. E7 always appears as a minor key enabler. There are not enablers classified as independent for MSEs and MSMEs.

## 4. Discussion

The main aspects of barriers and enablers for the integration of I4.0 and sustainability in supply chains of MSMEs are discussed: the prominent, influencing and resulting barriers and enablers, in a manner consistent with [53]. The reciprocal influences among barriers and among enablers, based on consensus rating of verbal scales obtained during the focus groups is presented first. Then, the section discusses the prominent, influencing and resulting factors extracted from the Fuzzy-DEMATEL approach of cause-effect relationships ensue. Research propositions are derived separately for MSEs and MEs.

### 4.1. Consensual Rating of Verbal Scales

Figures 4–7 present 8 × 8 matrices based on the academic literature and on empirical insights from practitioners, depicting the ratings of verbal scales during focus group discussions. They offer important highlights. First, no zero score was applied by any specialist, in any focus group, which means that the experts conclude that no barrier or enabler fails to influence each other. Second, no score was appointed either as fully or solely influencing the other. They are all intertwined. Third, when looking at the interrelationships in the subgroup of barriers and in the subgroup of enablers, rated by consensus for at least four out of six participants, a consensual picture emerges leading to four research propositions. Figure 14 depicts the barriers for MSEs and for MEs, separately.

Among MSEs, the lack of technical expertise very highly influences the barriers related to cybersecurity, resistances to change, lack of investment and perceived high costs

of improvement. Costs of improvement and the lack of commitment from top management raise other barriers. Consequently, the following proposition is put forward:

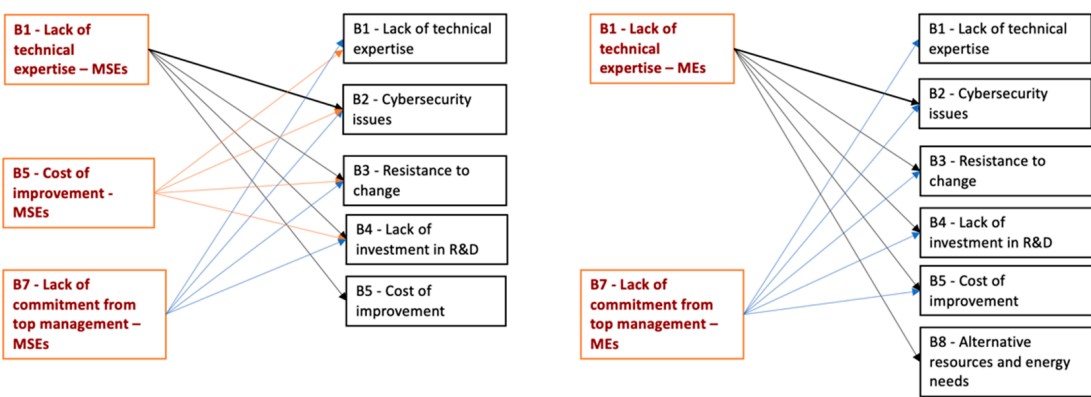

**Figure 14.** Main influencers of barriers for MSEs (B1; B5; B7) and MEs (B1; B7)—Focus Groups.

P1—In MSEs, the costs of improvements, the perceived lack of technical expertise and commitment from top management reinforce each other and trigger the barriers of cybersecurity issues, resistances to change, and lack of investments in R&D.

In MEs, it appears that the cost of improvement is less prevalent as a barrier than in MSEs. However, the lack of technical expertise and top-level commitment remains as major influencers of other barriers. Another distinguishing feature is the emergence of the barrier of the need for alternative resources and energy sources. In this context, it is plausible to advance the following proposition.

P2—In MEs, the lack of expertise and top-management commitment reinforce each other, triggering the barriers of cybersecurity issues, resistances to change, lack of investments in R&D, improvement costs and the need for alternative resources and energy sources.

The enablers for MSEs and MEs appear in Figure 15. In the case of MSEs, top management commitment and data centered solutions with consistent data flows would reinforce other enablers. Top-level commitment enhances employees' empowerment, innovation, a data driven culture and interdisciplinarity. Data driven solutions seems to burst top-level commitment, internal innovation, integration with customers and suppliers, and valuing R&D and research centers.

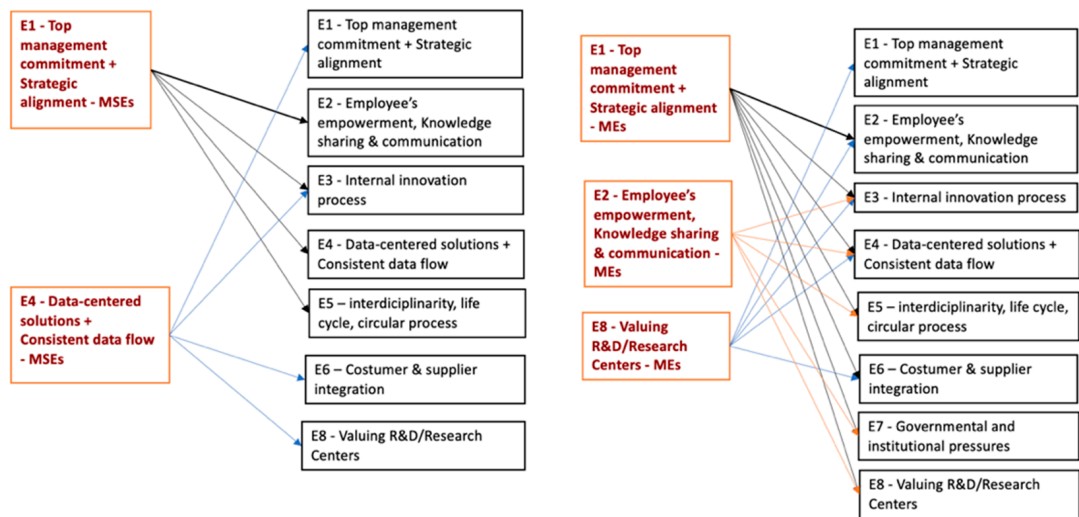

**Figure 15.** Main influencers of enablers for MSEs (E1; E4) and MEs (E1; E2; E8)—Focus Groups.

The following propositions ensue:

P3—In MSEs, top management commitment, strategic alignment and a data centered culture are mutually reinforcing mechanisms and serve as facilitators to the emergence of other enablers.

P4—In MSEs, top-management commitment and strategic alignment facilitate employees' empowerment, a data driven culture, internal innovation, interdisciplinarity and the adoption of life cycle and circularity processes.

P5—In MSEs, a data driven culture facilitates top-management commitment, strategic alignment, internal innovation, and integration with customers and clients.

In the case of MEs, top management commitment, strategic alignment, employees' empowerment, knowledge sharing and communication, and valuing R&D and research centers are the major enablers, triggering all eight consequents. Employees' empowerment does not appear as an enabler for MSEs as expected, most likely due to size. But it is a decisive factor among MEs. The following research propositions ensue:

P6—In MEs, top management commitment, strategic alignment, employees' empowerment and communication, R&D and research centers are mutually reinforcing mechanisms facilitating I4.0 and sustainability integration.

P7—In MEs, top-management commitment and strategic alignment are catalyzers of empowerment, internal innovation, data centered culture and integration.

P8—In MEs, employees' empowerment triggers internal innovation, a data centered culture, interdisciplinarity, R&D and research centers.

P9—In MEs, valuing R&D and research centers facilitate the emergence of top-management support and strategic alignment, empowerment, internal innovation, data-centered culture, and integration with customers and suppliers.

### 4.2. Prominence (D + R), Importance (D − R), and DEMATEL Map Discussion

The emerging key barriers and enablers are evaluated using the Fuzzy-DEMATEL method, and it helps to gain novel comprehensions with theoretical advances and practical implications. "D + R" score signifies the total cause and effect (prominence). The higher the value of the "D + R" the greater the influence of the barrier or enabler "y". The comparative ranking of prominent factors based on the "D + R" scores appears in Table 13, separately for MSEs, MEs and for both combined. The value of "D − R" shows the net effect of the barrier or enabler "y". The barrier or the enabler will be considered a cause (if positive) or a resulting (if negative) net effect (D − R). Consistent with [57], net effects can guide practical propositions as suggested in this paper. Companies would gain to focus on cause effects first and to attack resulting or consequent effects after [53], p.13. The most influential (D − R positive) and consequent factors (D − R negative) are different according to size. This fact triggers separate propositions for practice and policymaking according to enterprise size. Barriers or enablers (factors) with a higher D − R value generate greater influence, being considered of higher priority or importance [47].

**Table 13.** D + R Results for Barriers and Enablers.

| Factor | MSEs | MEs | MSMEs (Combined) |
| --- | --- | --- | --- |
| Barrier | B5 > B1 > B4 | B5 > B3 > B8 | B5 > B3 > B1 |
| Enablers | E4 > E5 > E6 | E8 > E1 > E2 | E8 > E2 > E1 |

Note: B1—Lack of technical expertise; B3—Resistance to change/change management practices and adopting innovation for society; B4—Lack of investment in R&D; B5—Cost of improvement & OSCM economic condition; B8—Alternative resources and energy needs. E1—Top management commitment + Strategic alignment; E2—Employee's empowerment + Knowledge sharing + Effective communication; E4—Data-centered solutions + Consistent data flow; E5—Interdisciplinary and holistic integration + Life cycle thinking and circular processes; E6—Customer and supplier integration; E8—Valuing R&D/Research Centers.

Once these cause factors (D − R values greater than 0) are calculated, Table 14 ranks barriers and enablers for MSEs, MEs and MSMEs. Indications of cause and effect are shown in Figures 8–13.

**Table 14.** D − R Results for Barriers and Enablers (values > 0).

| Factor | MSEs | MEs | MSMEs (Combined) |
|---|---|---|---|
| Barrier | B7 > B6 > B8 | B6 > B7 > B1 | B7 > B6 > B1 |
| Enablers | E7 > E1 > E2 | E1 > E7 | E7 > E1 > E2 |

Note: B1—Lack of technical expertise; B6—Lack of support from regulatory authority/poor legislation; B7—Lack of commitment from top management; B8—Alternative resources and energy needs. E1—Top management commitment + Strategic alignment; E2—Employee's empowerment + Knowledge sharing + Effective communication; E7—Governmental and institutional pressures; E8—Valuing R&D/Research Centers.

The magnitude and ordering of factors vary according to firm size, which leads to two separate research propositions, as follows:

P10—In MSEs, the main roadblocks of costs of R&D, improvements and expertise could be circumvented or mitigated by a data centered culture, with interdisciplinary integration, internal and in the supply chain, allied with the embracement of life cycle and circularity processes.

P11—In MEs, the impediments of improvement costs, resistances to change and the need for resources and alternative sources of energy can be mitigated by the prominent enablers of top-management support, strategic alignment, employees' empowerment, knowledge sharing and communications.

Finally, four propositions for practitioners and policy-makers are offered to complete the discussions an interpretation of the research findings.

P12—MSEs should first address the cause barriers of lack of commitment from top management (B7), lack of support from regulatory authority/poor legislation (B6), the need for alternative resources and energy sources (B8), and the lack of expertise (B1). The remaining barriers should be addressed sequentially.

P13—MSEs would benefit from priority investments of time and efforts in the cause enablers of governmental and institutional support (E7), top management commitment and strategic alignment (E1), and Employee's empowerment, knowledge sharing, and effective communication (E2). The remaining enablers could be contemplated as a second order priority.

P14—MSMEs priority actions to circumvent cause barriers should gear at mitigating the lack of support from regulatory authority/poor legislation (B6), the lack of commitment from top management (B7), and the lack of technical expertise (B1). The consequent barriers should be tackled later. B1, B6 and B7 are key barriers and should receive priority in the treatment by the company and by other stakeholders involved, such as governments and educational institutions and technical training. The Alternative resources and energy needs (B8) barrier also draws attention as key barriers for MSEs and as an indirect barrier for MEs. In the combined MSMEs predominates as a key barrier, which allows us to attest greater difficulty for MSEs to overcome this barrier due to less formal access to clean energy financing programs, especially in developing countries. One possible suggestion is to reduce real financial guarantees so that MSEs can access, in scale, the financing available or that may become available.

P15—Policy and action programs for MEs should be directed in priority to enhance the cause enablers of top management commitment and strategic alignment (E1), and to burst governmental and institutional support (E7), before addressing consequential enablers. The enabler Top management commitment + Strategic alignment (E1) is considered a key enabler. Another highlight is Employee's empowerment + Knowledge sharing + Effective communication (E2) considered indirect enabler for MEs and key enabler for MSEs and combined MSMEs. This result makes a lot of sense when looking at key barriers because these two enablers are directly associated mainly with key barriers B1 and B7.

## 5. Conclusions and Future Research

Although the integration of I4.0 and sustainability in supply chains has been considered by academics and practitioners from the industry, there is a scarcity of studies on

this integration from a MSMEs perspective. This gap results in a research opportunity, as solutions for large sized enterprises cannot necessary be generalized for MSMEs, who play an important role in the economy of developing countries and in supply chains of large sized enterprises. This paper addresses this research gap by identifying main barriers and enablers for the integration of I4.0 and sustainability in supply chains of MSMEs, as well as their resulting causalities, dependencies, and hierarchical levels. A detailed multimethod approach is taken, resulting in eleven research propositions and four propositions for practitioners and policymakers.

Starting from a list of 13 barriers and 31 enablers obtained from the scoping review, eight main barriers and enablers geared to the reality of MSMEs emerged through the panel of experts. Their respective cause-effect relationships were revealed within two focus groups applying the Fuzzy-DEMATEL method. The separation of MSMEs into two groups with specialists segregated in MSEs and MEs proved to be useful as research findings show significant differences from interviews and focus groups, what sheds new lights in the existing heterogeneity within MSMEs, which deserve attention in future research. The methodology used allowed for an analysis between the results of the groups of experts and the direct comparison of each group with the grouped result. Barriers and enablers are positioned differently within the graphs and often drop out of the cause group and appear in the effect group, and vice-versa. The identification of cause groups and effect groups for barriers and enablers is extremely important for MSMEs, as it allows them to concentrate greater efforts on barriers and enablers belonging to the cause group. Therefore, findings are expected to help MSMEs managers and decision makers to better understand and implement the integration I4.0 and sustainability in their supply chains. Despite the contextual differences and different orders of magnitudes for cause barriers and cause enablers between MSEs and MEs, the content of actions to mitigate barriers and promote enablers have some commonalities. Regarding barriers, actions should gear in priority to the circumvention of the lack of commitment from top management, support from regulatory authority/poor legislation, lack of expertise, and the need for alternative resources and energy sources. Potentially rewarding enablers in common to MSEs and MEs are the need to enhance top management commitment and strategic alignment, and to burst governmental and institutional support.

The results, discussions, and propositions of this research offer additional practical opportunities for managers' reflection. It is challenging to look at the implementation of I4.0 technologies with a sustainable perspective on the supply chain. The interaction with the environment outside the industry is an additional challenge. In the proposals, there are examples that can directly help managers, such as: (i) implementation of policies; (ii) establishment of action programs; (iii) identification of key factors, prioritizing the treatment of barriers and enablers; (iv) offer of concrete arguments for dealing with other interest groups involved such as governments, as in the case of public policies to encourage the use of clean energy sources.

As the implementation of I4.0 within sustainability in supply chains is still recent in literature and associated to constant transformation and evolution of technologies, the topic should be further explored and investigated through additional empirical research. The results are most likely to be different depending on the analyzed context, as the degree of technological maturity of the enterprise or the market/region in which the enterprise operates varies, which might be better viewed through the lenses of contingency theory e.g., [58]. Another opportunity may lie in MSMEs segmentation (e.g., MSEs and MEs), for instance, by market or intensity of I4.0 technologies, that allows a greater deepening of their impact on sustainability of the company, in economic, environmental, and social pillars. Technology-intensive sectors or experts who have experience in companies in these sectors may present different perceptions and analyses. A focus group with experts with experience in technology-intensive markets can allow market, regional or relational comparisons, among others. Finally, future research can also explore the research limitations of this paper. The scoping review was broad and general by design as a first research step of the adopted

convergent parallel mixed method approach. This first did not aim to a balance the number of barriers and enablers, but to bring the actual state of the art in the extant literature, to be further examined and refined by the panel of judges and focus groups discussions towards a MSME perspective. Moreover, the scoping review returned a limited number of studies. Within this context, a systematic literature review is recommended as future research to overcome these limitations offering not only a broader inclusion criterion to retrieve more papers, but also framework and taxonomy as recommended in [32]. As the interviews were limited to experts from Brazil, other emerging countries as well as developed countries should be considered further to extend the generalization of this study's findings and to verify possible differences between emerging and developing countries.

**Author Contributions:** Conceptualization, E.M., R.G.G.C. and L.F.S.; methodology, E.M., R.G.G.C., L.F.S. and A.M.T.T.; software, E.M., R.G.G.C., L.F.S. and A.M.T.T.; validation, E.M. and R.G.G.C.; formal analysis, E.M. and R.G.G.C.; investigation, E.M., R.G.G.C. and L.F.S.; resources, E.M., R.G.G.C., L.F.S. and A.M.T.T.; data curation, E.M.; writing—original draft preparation, E.M., R.G.G.C., L.F.S. and A.M.T.T.; writing—review and editing, E.M., R.G.G.C., L.F.S. and A.M.T.T.; visualization, E.M.; supervision, R.G.G.C. and L.F.S.; project administration, E.M., R.G.G.C., L.F.S. and A.M.T.T.; funding acquisition, R.G.G.C., L.F.S. and A.M.T.T. All authors have read and agreed to the published version of the manuscript.

**Funding:** This work was supported by the Coordenação de Aperfeiçoamento de Pessoal de Nível Superior—Brazil—CAPES (Finance Code 001), the Brazilian National Council for Scientific and Technological Development—CNPq (grant number 311757/2018-9, 311862/2019-5; 404682/2016-2; 304931/2016-0), and the Carlos Chagas Filho Foundation for Research Support of the State of Rio de Janeiro—FAPERJ (Grant Numbers E-26/203.252/2017; E-26/201.251/2021; E-26/201.363/2021).

**Institutional Review Board Statement:** Not applicable.

**Informed Consent Statement:** Not applicable.

**Acknowledgments:** The authors would like to thank all the experts who also supported this research.

**Conflicts of Interest:** The authors declare no conflict of interest.

## Appendix A

QUESTIONNAIRE (First round—Panel of Experts)

Part 1—Research on barriers and enablers for the adoption of Industry 4.0 technologies (I4.0) in sustainable supply chains of micro, small and medium enterprises (MSMEs)

FREE AND INFORMED CONSENT FORM

Responsible: PhD Student: Eduardo Augusto Machado, and Professors: D.Sc. Luiz Felipe Scavarda/D.Sc. Rodrigo Caiado

You are being invited to volunteer in a survey/interview. This document, called the Informed Consent Term (ICT), is intended to ensure your rights as a participant and you can keep a copy of it if you wish. Please read carefully and calmly, taking the opportunity to clarify your doubts. If there are questions before or even after indicating your agreement electronically, you can clarify them with the researchers at the time of the survey, in person or online. There will be no damages if you do not accept to participate or withdraw your authorization at any time. Your identification will not be used.

Objectives: The main justification for conducting this research/interview is to validate barriers and enablers for the adoption of Industry 4.0 technologies in sustainable supply chains of micro, small and medium enterprises (MSMEs). Both barriers and enablers will be presented to you, who in turn must respond indicating the intensity of each item in relation to its respective importance for MSMEs. At the end of each of the study parameters (barriers and enablers), you can freely indicate new barriers and enablers that are not yet considered, and you understand that should be listed.

Procedures: Initially, check the first option stating that you want to participate as a volunteer. Then, enter your data and your email if you wish to receive the results statistically treated and a final report of the survey. Please answer the questions about

barriers and enablers based on your knowledge and experience. In the following sections, assess indicating the degree of intensity of barriers and enablers in terms of importance and influence for MSMEs.

Glossary:

- Barrier: Resistant force that makes it difficult to carry out activities [55].
- Enabler: Also characterized as an inducer, a critical success factor. Critical success factors can be understood as organizational actions necessary to ensure success and competitiveness, supporting a company's organizational change processes [59].
- Industry 4.0 (I4.0): Trend towards digitization and automation of the manufacturing environment [60], confluence of technologies from a variety of digital technologies [61], new stage or paradigm of industrial production, focusing on the results of transformation process [62]. It aims to link disruptive technologies to manufacturing systems, combining intelligent operations and supply chain management (OSCM) [1].
- I4.0 Technologies: IoT (Internet of Things); CPS (Cyber Physical Systems); BDA (Big Data Analytics); Cybersecurity; Cloud Computing; AM (Additive Manufacturing)/3DP (3D Printing), AR (Augmented Reality)/VR (Virtual Reality); Advanced Robotics; Blockchain.
- Sustainable supply chain: Management of materials, information, and capital flows, as well as cooperation between companies along the supply chain, having goals from all three dimensions of sustainable development (economic, environmental, and social) [63].
- Developing countries: Very heterogeneous concept, the designations "developed" and "developing" are intended for statistical convenience and do not necessarily express a judgment about the stage reached by a particular country or region in the development process. Generally speaking, there are countries with a Human Development Index (HDI) between 0.555 and 0.799 [64–66].
- Micro enterprise: Annual turnover equal to or less than R$360,000.00 (US$72,000.00—parity US$ 1 = R$5), also characterized as companies with up to 9 (nine) employees [67].
- Small company: Annual revenue equal to or less than R$4,800,000.00 and more than R$360,000.00 (less than US$ 960,000.00 and more than US$72,000.00—parity US$ 1 = R$5), also characterized as companies with 10 to 49 employees (services and commerce) and between 20 and 99 employees (industry) [67].
- Medium company: Annual turnover equal to or less than R$ 20,000,000.00 and greater than R$4,800,000.00 (less than US$ 5,000,000.00 and more than US$960,000.00—parity US$1 = R$5) also characterized as companies with 50 to 99 employees (services and commerce) and between 100 and 499 employees (industry) [67].

Do you accept to participate as a volunteer in this research?

(_) Yes/(_) No

Please enter your e-mail address if you would like to receive the survey results:

______________________________

Part 2—Profile

What is your accumulated time (in years) of experience with or in MSMEs?_________
What size of company is your accumulated experience with or in MSMEs?

(_) Micro and small companies
(_) Medium companies
(_) All (micro, small, and medium companies)

What is your knowledge/experience with I4.0 technologies?

(_) None
(_) Up to 2 years
(_) From 3 to 5 years
(_) Over 5 years

What is your knowledge/experience with sustainability in supply chains?

(_) None
(_) Up to 4 years
(_) From 5 to 9 years
(_) Over 10 years

Part 3 BARRIERS—This section lists barriers to deploying Industry 4.0 (I4.0) technologies in sustainable supply chains. The objective is: to indicate the degree of importance of each of these barriers for MSMEs.

(_) None; (_) Very Low; (_) Low; (_) Medium; (_) High; (_) Very High; (_) Fully

Cybersecurity issues
Cost of improvement & OSCM economic condition
Lack of support from regulatory authority/poor legislation
Lack of commitment from top management
Interoperability issues
Employability
Lack of technical expertise
Alternative resources and energy needs
Design challenges to reuse and recovery products
Organizational barriers, Capacity constraints
Resistance to change/change management practices and adopting innovation for society
Data management and storage issues
Inequalities of opportunities and digitalization risks

Are there any BARRIERS you want to add that are not listed? (If in your opinion the list is complete, just answer 'no', if you believe there are one or more BARRIERS that should be included in the list, just write in the sequence you want:_____________

Part 4 ENABLERS—This section lists enablers for deploying Industry 4.0 (I4.0) technologies in sustainable supply chains. The aim is for you to indicate how important each of these facilitators is to MSMEs.

(_) None; (_) Very Low; (_) Low; (_) Medium; (_) High; (_) Very High; (_) Fully

Internal innovation process
Open innovation
Change management
Dynamic capabilities
Innovative business models and service design
Customer and supplier integration
Support of unconventional partners
Governmental and institutional pressures
Collaborative networks
Innovative business models and service design techniques
Re-designing and decentralized structure
Strategic alignment
I4.0 readiness
Adoption of smart factory components
Data-centered solutions
Consistent data flow
Modular design
Information transparency and data security
Sustainable philosophy
Focus on renewable natural resources
Interdisciplinary and holistic integration
Sharing economy
Life cycle thinking and circular processes
Knowledge sharing

Effective communication
Individual incentive schemes
Employee's empowerment
Experimentation
Education and training focused on soft and technical skills
Transformational leadership
Top management commitment)

Are there ENABLERS you want to add that are not listed? (If in your opinion the list is complete, just answer 'no', if you believe there are one or more ENABLERS that should be included in the list, just write in the sequence you want: __________

**Appendix B**

QUESTIONNAIRE (Second round—Panel of Experts)

Part 1—Complementary part of the first round of the panel with experts from on barriers and enablers to the adoption of Industry 4.0 technologies (I4.0) in sustainable supply chains of micro, small and medium enterprises (MSMEs)

Complementary research with experts

Responsible: PhD Student: Eduardo Augusto Machado, and Professors: PhD Luiz Felipe Scavarda/PhD Rodrigo Caiado

Dear expert, your questionnaire was considered complete and met all validation requirements. During the data collection process of the questionnaire, there were filtered out new 3 (three) BARRIERS and new 2 (two) ENABLERS. We need you to assess the relevance of these 3 BARRIERS and 2 ENABLERS by answering the 2 extra sections of this research complement. This complementary questionnaire is quick to complete, and we thank you for your attention.

Note: The "informed consent terms (ICT)" you agreed to in the main questionnaire are strictly maintained.

Part 2—Accept/expert email

Do you accept to participate as a volunteer in this research complement?

(_) Yes/(_) No. Indicate your email: ___________

Part 3—BARRIERS

This section presents barriers that were filtered out from the nominations in the open-ended question for that question.

There were 3 extra BARRIERS for your assessment:

- Cultural aspects
- Sub-utilization of academic knowledge/universities
- Lack of investment in R&D

There will be 2 questions for each BARRIER, if you as an expert understand that:

(a) Is it really a BARRIER for the implementation of Industry 4.0 technologies (I4.0) in sustainable supply chains of micro, small and medium enterprises (MSMEs)?
(b) Indicate the degree of importance of each extra barrier for MSMEs?

Do you consider 'Cultural aspects/characteristics' a barrier to the implementation of Industry 4.0 (I4.0) technologies in sustainable supply chains for micro, small and medium enterprises (MSMEs)?

(_) Yes/(_) No

If you consider 'Cultural aspects' a barrier, please indicate how important it is to MSMEs. NOTE: If you don't CONSIDER (marked 'NO' in the previous question, mark the option NONE):

(_) None; (_) Very Low; (_) Low; (_) Medium; (_) High; (_) Very High; (_) Fully

Do you consider 'Sub-utilization of academic knowledge/universities' a barrier to the implementation of Industry 4.0 (I4.0) technologies in sustainable supply chains of micro, small and medium-sized companies (MSMEs)?

(_) Yes/(_) No

If you consider 'Sub-utilization of academic knowledge/universities' a barrier, please indicate how important it is to MSMEs. NOTE: If 'NO' CONSIDERS (you checked no in the previous question, check the option NONE):

(_) None; (_) Very Low; (_) Low; (_) Medium; (_) High; (_) Very High; (_) Fully

Do you consider 'Lack of investment in R&D (Lack of investment in R&D)' a barrier to the implementation of Industry 4.0 (I4.0) technologies in sustainable supply chains of micro, small and medium enterprises (MSMEs)?

(_) Yes/(_) No

If you consider 'Lack of investment in R&D (Lack of investment in R&D)' a barrier, indicate how important it is for MSMEs. NOTE: If you don't CONSIDER (marked 'NO' in the previous question, mark the option NONE):

(_) None; (_) Very Low; (_) Low; (_) Medium; (_) High; (_) Very High; (_) Fully

Part 4—ENABLERS

This section features facilitators that have been filtered out of the nominations in the open-ended question for that question.

There were 2 extra ENABLERS for your evaluation:

- Better management of Certifications, Standards and Regulations
- Valuing R&D/Research Centers

There will be 2 questions for each ENABLER, if you as an expert understand that:

(a) Is it really an ENABLER for the implementation of Industry 4.0 technologies (I4.0) in sustainable supply chains of micro, small and medium enterprises (MSMEs)?
(b) Indicate the degree of importance of each extra facilitator to MSMEs?

Do you consider 'Better Management of Certifications, Standards and Regulations' an enabler for the implementation of Industry 4.0 (I4.0) technologies in sustainable supply chains of micro, small and medium enterprises (MPMEs)?

(_) Yes/(_) No

If you consider 'Better Management of Certifications, Standards and Regulations' an enabler, please indicate how important it is to MSMEs. NOTE: If you don't CONSIDER (marked 'NO' in the previous question, mark the option NONE):

(_) None; (_) Very Low; (_) Low; (_) Medium; (_) High; (_) Very High; (_) Fully

Do you consider 'Valuing R&D/Research Centers' an enabler for the implementation of Industry 4.0 (I4.0) technologies in sustainable supply chains of micro, small and medium enterprises (MSMEs)?

(_) Yes/(_) No

If you consider 'Valuing R&D/Research Centers' an enabler, please indicate how important it is to MSMEs. NOTE: If you don't CONSIDER (marked 'NO' in the previous question, mark the option NONE):

(_) None; (_) Very Low; (_) Low; (_) Medium; (_) High; (_) Very High; (_) Fully

**Appendix C**

DEMATEL methodology: There are different approaches to Fuzzy-DEMATEL. In this research, it was used for the scale, and also as a form of aggregation with triangular distribution for two groups (micro and small companies, and medium companies). Then,

the fuzzification is carried out considering the responses of these two groups. After this step, defuzzification occurs, where values are generated that indicate which factors (barriers and enablers) are cause or effect. The aggregation of the two groups generates a third aggregated group that uses the average of the appropriation values obtained previously (Tables 10 and 11, Figures 12 and 13).

(C1) Create the correspondence of linguistic terms and values (Table 2)

(C2) Aggregate results and obtain a fuzzy pairwise direct-relation matrix ($X$):

A composite survey instrument based on peer comparisons of barriers is completed by experts. Expert assessment is added by calculating average scores and forming aggregated direct relationship matrices. When the number of factors is n, the pair comparison matrix, $X$, is n × n. Each element within this matrix, $Xij$, represents the level of influence of factor *i* on a factor *j*. The influence of each factor on itself that forms the diagonal of the direct relation matrix is nullified. The general matrix of direct pair relationship is presented, following [40]:

$$\widetilde{z} = \frac{\left(\widetilde{z}^1 \oplus \widetilde{z}^2 \oplus \cdots \oplus \widetilde{z}^p\right)}{p}$$

$$\widetilde{Z} = \begin{bmatrix} \widetilde{z}_{11} & \widetilde{z}_{12} & \cdots & \widetilde{z}_{1n} \\ \widetilde{z}_{21} & \widetilde{z}_{21} & 0 & \widetilde{z}_{2n} \\ \vdots & \vdots & \ddots & \vdots \\ \widetilde{z}_{n1} & \widetilde{z}_{n2} & \cdots & \widetilde{z}_{nn} \end{bmatrix}$$

Relation Fuzzy matrix $X$ by using

$$\widetilde{X} = \begin{bmatrix} \widetilde{x}_{11} & \widetilde{\chi}_{12} & \cdots & \widetilde{x}_{1n} \\ \widetilde{x}_{21} & \widetilde{x}_{21} & 0 & \widetilde{x}_{2n} \\ \vdots & \vdots & \ddots & \vdots \\ \widetilde{x}_{n1} & \widetilde{X}_{n2} & \cdots & \widetilde{x}_{nn} \end{bmatrix}$$

where:

$$\widetilde{x}_{ij} = \frac{\widetilde{z}_{ij}}{r} = \left(\frac{l_{ij}}{r}, \frac{m_{ij}}{r}, \frac{n_{ij}}{r}\right)$$

$$r = \max_{k \leq i \leq n}\left(\sum_{j=1}^{n} u_{ij}\right).$$

It is assumed at least one *i* such that $\sum_{j=1}^{n} u_{ij} < r$.

(C3) Normalize the direct-relation matrix and calculate the total relation matrix (T) that determines the relationship between factors where *I* is the identity matrix:

After computing the above matrices, the total-relation fuzzy matrix $\widetilde{T}$ is computed. Total-relation fuzzy matrix is defined as: [48]

$$\widetilde{T} = \lim_{k \to \infty}\left(\widetilde{X}^1 + \widetilde{X}^2 + \cdots \widetilde{X}^k\right)$$

then,

$$\widetilde{T} = \begin{bmatrix} \widetilde{t}_{11} & \widetilde{t}_{12} & \cdots & \widetilde{t}_{1n} \\ \widetilde{t}_{21} & \widetilde{t}_{21} & 0 & \widetilde{t}_{2n} \\ \vdots & \vdots & \ddots & \vdots \\ \widetilde{t}_{n1} & \widetilde{t}_{n2} & \cdots & \widetilde{t}_{nn} \end{bmatrix}$$

In which $\widetilde{t}_{ij} = \left(l''_{ij}, m''_{ij}, u''_{ij}\right)$ and

$$\left[l''_{ij}\right] = X_l \times \left(I - X_1^{-1}\right), \left[m''_{ij}\right] = X_l \times \left(I - X_m^{-1}\right), \left[u''_{ij}\right]$$
$$= X_l \times \left(I - X_u^{-1}\right)$$

(C4) Determine row and column sums from *T*:

Given that *Tij* is the comparison variable of the factor *i* on the factor j in the total relation matrix, *T*, where *i,j* =1, 2, ..., n, the row (*Di*) and column (*Rj*) sum for each row *i* and column *j* are obtained using expressions:

$$D_i = \sum_{j=1}^{n} t_{ij} \; \forall i$$

$$R_j = \sum_{i=1}^{n} t_{ij} \; \forall j$$

By producing matrix $\widetilde{T}$, then it is calculated $\widetilde{D}_i + \widetilde{R}_j$ and $\widetilde{D}_i - \widetilde{R}_j$ in which $\widetilde{D}_i$ and $\widetilde{R}_j$ are the sum of row and the sum of columns of $\widetilde{T}$ respectively. To finalize the procedure, all calculated $\widetilde{D}_i + \widetilde{R}_j$ and $\widetilde{D}_i - Rj$ are defuzified through suitable defuzification method. Then, there would be two sets of numbers: $\left(\widetilde{D}_i + Rj\right)^{def}$ which shows how important the strategic objectives are, and $\left(\widetilde{D}_i - \widetilde{R}_j\right)^{def}$ which shows which strategic objective is cause and which one is effect. Generally, if the value $\left(\widetilde{D}_i - \widetilde{R}_j\right)^{def}$ is positive, the objectives belong to the cause group, and if the value $\left(\widetilde{D}_i - \widetilde{R}_j\right)^{def}$ is negative, the objectives belong to the effect group.

(C5) Determine the overall prominence and net effect values of factors:

The overall value by which a factor is being influenced and its influence on other factors characterizes overall prominence (*P*). The difference between the impact that a factor has on others and the impact received by others characterizes the net effect value (*Ei*). *Pi* and *Ei* can be calculated by the expressions:

$$Pi = \{Di + Rj \mid i = j\}$$

$$Ei = \{Di - Rj \mid i = j\}$$

(C6) Formulate the DEMATEL cause-effect Diagrams:

The last step is the graphical representation for each factor (barriers and enablers) of the calculated prominence and net effect values on a two-dimensional axis. *X*-axis is the prominence value; the *y*-axis is the net effect value. The threshold value θ [68] is defined by the expression:

$$\theta = mean\ (T) + SDT$$

**Appendix D**

Figures A1 and A2 present, respectively, the evaluation of the Panel of Experts for the 16 barriers and 33 enablers.

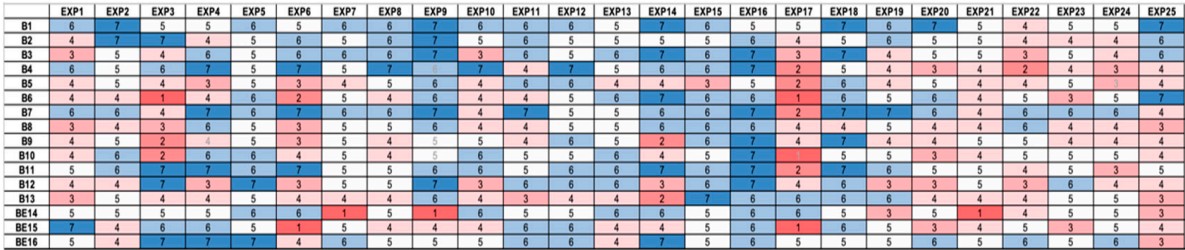

**Figure A1.** 25 Experts values for each barrier.

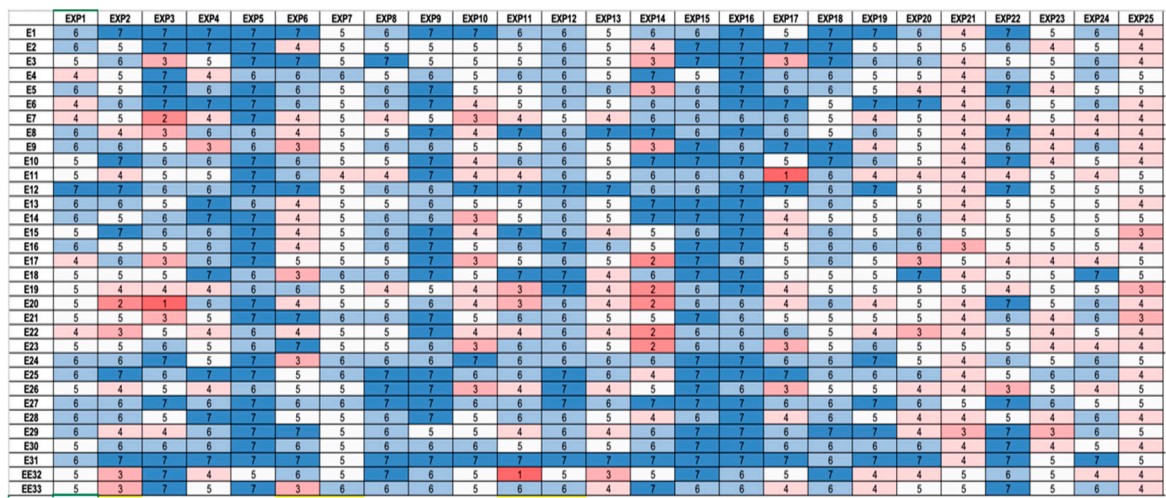

**Figure A2.** 25 Experts values for each enabler.

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
