# Peer review of "Barriers and Enablers for the Integration of Industry 4.0 and Sustainability in Supply Chains of MSMEs"

_sustainability, doi:10.3390/su132111664_

Round 1
Reviewer 1 Report
I consider the paper to be of high quality.
There are two issues that could be improved. First, the justification and explanation of the passage from step 1 to step 2, and step 2 to step 3 is not clear enough. That could be improved. We understand the importance of barriers and enablers. But how and why do you pass from a scoping review to a panel of experts and then to a focus group.
I would suggest that a better justification or explanation of each step and why at the end is the next step necessary and useful in the demonstration. For me the transition from one step to another is not that clear. Obviously the technique used (fuzzy Dematel) is rather clear, but why to use this technique.
In this line, the results of the scoping review could be biased, because of the limited number of studies that are scattered around different sectors. This can limit the broadness of the results. As a matter of a fact, there are much more enablers than barriers.
The second aspect is the consideration of the context of each study that is retained in the scoping review. Many other contexts could suggest other enablers and barriers. But even with similar enablers and barriers, their importance could be discutable. The experts are aware of one national context. Brazil. Are they others? You mention something about it in the last part of the discussion (line 501).
The scoping review is not focused more on industrialized countries with an environmental legislation and institutional apparatus that is different from the one that the interviewers are familiar with?
The third aspect is the characterization of enablers and barriers. As I said, why much more enablers than barriers? What about the bias possible in the literature and the resulting scoping review. But the question here has much to do with the categories of the enablers and barriers. Are they symmetrical? On table 3, E3 and B10 and B11 seem to fit in a similar category.; the same between E31 and B4 or other ones.
In table 4 why are categories for barriers and enablers related? Why “social” for barriers and “people” for enablers. Some are the same like innovation, technology, legal, etc. But others are not so clear how they fit. For example, why institutional in the categories of enablers? Why a category of environment in barriers and one of sustainability in enablers? In table 6, the category Social as barrier and people in enablers are related or distinguishable?
Some minor points or questions, now.
In the tables 7 and after, the interpretation for Di+Ri is rather straightforward. But it is not the case for Di-Ri.
In several parts of the paper, the reference made to authors with numbers are used as subjects of the sentences, grammatically speaking. I find that odd. See line 35, 68, 6970, 74, 86, 90, 96, etc. I would expect the names of the author or first two or three.
The conclusion is good but could be improved in terms of interpretation of the results. What does it mean for industrial managers that focus on sustainability or even industry 4?

Author Response
REVIEWER#1: COMMENTS & SUGGESTIONS FOR AUTHORS
Reviewer #1: I consider the paper to be of high quality.
Authors: We appreciate very much your comment about our paper. Indeed, being our paper of “high quality” motivates us to continue to work hard on our research.
Reviewer #1: There are two issues that could be improved. First, the justification and explanation of the passage from step 1 to step 2, and step 2 to step 3 is not clear enough. That could be improved. We understand the importance of barriers and enablers. But how and why do you pass from a scoping review to a panel of experts and then to a focus group. I would suggest that a better justification or explanation of each step and why at the end is the next step necessary and useful in the demonstration. For me the transition from one step to another is not that clear.
Authors: Thanks for your comment and nice suggestion! We changed different parts of the manuscript including addition justifications and explanations to address this important concern, with which we do also share. Please find next a description of the main changes conducted in the manuscript.
1) At the end of the introduction section, just after the research questions and goals, we provide addition explanations for each of the three steps to prepare the reader for the multimethod approach adopted with these three steps. We copy below part of this paragraph highlighting the new content in blue to ease your review.
“.... To this end, a multimethod approach is adopted by combining a scoping review to identify general barriers, enablers, and associated categories available in the literature, a panel of experts with two rounds to refine and classify the identified barriers and enablers towards the perspective of MSMEs, and finally two focus groups using the fuzzy logic and DEMATEL methods to obtain the inter-relationship of the main barriers and enablers for MSMEs. As a result, this study makes manifold contributions to the literature on this topic by …”.
2) In the first paragraph of Section 2 (Materials and Methods), the rationale for transitioning from steps 1, 2 and 3 is now introduced, as follows. We highlight the new content in blue to ease your review.
“This research adopts a convergent parallel multimethod approach [27] based on three steps: (i) scoping review, (ii) panel of experts, and (iii) focus groups, as shown in Figure 1. In convergent parallel multimethod designs, the researcher combines almost simultaneously qualitative and quantitative methods, triangulating methods and data in the search of a deeper understanding of the phenomenon under study. The broad categories for the identification of enablers and barriers were identified in step 1, during the scoping review. Sequentially, a panel of experts was convened to refine the typology and to identify antecedents and consequents to rank barriers and enablers identified in phase 1. The focus group took place sequentially after step2 to clarify issues of consensus and disagreements among judges. Each step is detailed throughout this section two. The rationale to transition from one step to the next is explained for each step below.”
3) In addition, the following explanatory paragraphs were added to each step, as transcribed next.
3.1) At the subsection 2.1 (Scoping review) - End of the first paragraph. As it is completely a new content, it is all in blue.
“ … According to [30], scoping is more general than systematic reviews by design: “a key difference between scoping reviews and systematic reviews is that in terms of a review question, a scoping review will have a broader “scope” than traditional systematic reviews with correspondingly more expansive inclusion criteria.” Therefore, this research step aims to scan the literature available and then picks out the barriers and enablers from work already carried out, as in [31]. Additionally, research gaps are identified. Thus, the gaps and, consequently, the research questions (derived from the gaps pointed out in section 1) that guided this study, are identified and sorted according to a scoping review [28]. The purpose of the scoping review is to identify and analyze knowledge gaps, to identify the types of available evidence in the field of I4.0 and SSCM, and to identify barriers and enablers (factors) [30], related to the concept of the integration of I4.0 and sustainability in supply chains of MSMEs.”
Note: In the Scoping Review phase (step 1), there was no geographic delimitation, nor any delimitation related to the size of the companies. The focus was on extracting barriers and facilitators from the literature related to the sustainable implementation of I4.0 technologies. The contemplation of MSMEs is covered by the conquest of specialists with proven experience in MSMEs (next step). As discussed in the introduction to the article, there is a lack of research aimed at affecting MSMEs supply chains for emerging countries where MSMEs are very representative for the economy. Brazil was covered by the survey due to its importance and representativeness for emerging countries. It is a research limitation, which certainly presents an opportunity to be addressed in future research, as highlighted at the end of the article.
3.2) At the subsection 2.2 Panel of Experts - End of the first paragraph. As it is completely a new content, it is all in blue.
“… On the expert panel, anonymous responses are distributed to experts, who are allowed to review their own responses in subsequent rounds until consensus is reached [39]. Based on the perception of specialists in MSMEs, this steps seeks to filter and refine barriers and facilitators (factors), where the main ones will be identified [40]. This step is supported by experts to validate the list of barriers and facilitators (factors) and to find other factors based on professional views and experiences [31]. Thus, the identified factors are complemented through expert panels [40], through a specific questionnaire aimed at grouping and selecting factors that adhere to the reality of MSMEs in a developing country.”
3.3) At the subsection 2.3 Focus Group - Beginning of the first paragraph. As it is completely a new content, it is all in blue.
“With focus groups, this study seeks consensus, considering focus groups as fundamental units of analysis through homogeneous participants and a moderator prepared to stimulate constructive discussion [45]. Focus groups are appropriate, as they allow questions aimed at assessing both the influence and priority of barriers and facilitators in the context of MSMEs. They also provide an environment for in-depth discussion to help formulate research proposals. Focus groups are similar to panel of experts, however, they are structured for verbal responses and exchanges rather than in writing. Thus, everyone in the group is aware of the origin of the answers. The group is given a set of questions, usually before the meeting. The facilitator asks the questions and allows each member to express their opinion. Discussion is allowed, stimulated, and controlled by the facilitator, always with the objective of obtaining consensus [39]. ...”
Reviewer #1: Obviously, the technique used (fuzzy Dematel) is rather clear, but why to use this technique.
Authors: Thank you for this comment and to provide us the opportunity to clarify better the use of the fuzzy Dematel. We improved our justification for its use in the second part of the subsection 2.3 focus group. The main inclusions are copied next to ease your review:
New paragraph after table 1 (paragraph 3):
“To quantify the relationship between evaluation factors, they give their linguistic assessments in the forms of intuitionistic fuzzy sets representing which factors have direct relation with each other [47]. The DEMATEL method identifies the cause-effect relationship of the criteria related to the problem, being effective to visualize the structure of complicated causal relationships with matrices or digraphs, which in turn portray a contextual relationship between the elements of the system, in which a numeral represents the force of influence, making the relationship structured and intelligible. To establish a structural model of experts' judgments, their respective preferences and importance are attributed with notable values, which in turn are inadequate in the real world as they are often obscure and difficult to estimate by exact numerical values, thus creating the need of fuzzy logic [48].”
Beginning of Paragraph 6:
“In this research, fuzzy set theory has also been embedded with DEMATEL to overcome the inaccuracy and bias of experts’ decisions [50]. The Fuzzy-DEMATEL method was applied to assess. The Fuzzy-DEMATEL method was applied to assess the causal relationship between barriers and enablers using expert input. According to [51], expert assessments contain inaccuracies and subjectivity, and the fuzzy theory directly addresses this issue as indicated by [52]. …”
Reviewer #1: In this line, the results of the scoping review could be biased, because of the limited number of studies that are scattered around different sectors. This can limit the broadness of the results. As a matter of a fact, there are much more enablers than barriers.
Authors: Thank you for noting the limitations of the scoping review. We address this point now at the end of the conclusion section (last paragraph). To ease your review, we copy next the new text.
“… Finally, future research can also explore the research limitations of this paper. The scoping review was broad and general by design as a first research step of the adopted convergent parallel mixed method approach. This first step did not aim to a balance the number of barriers and enablers, but to bring the actual state of the art in the extant literature, to be further examined and refined by the panel of judges and focus groups discussions towards a MSME perspective. Moreover, the scoping review returned a limited number of studies. Within this context, a systematic literature review is recommended as future research to overcome these limitations offering not only a broader inclusion criterion to retrieve more papers, but also framework and taxonomy as recommended in [32]. …”
Reviewer #1: The second aspect is the consideration of the context of each study that is retained in the scoping review. Many other contexts could suggest other enablers and barriers. But even with similar enablers and barriers, their importance could be discutable. The experts are aware of one national context. Brazil. Are they others? You mention something about it in the last part of the discussion (line 501).
The scoping review is not focused more on industrialized countries with an environmental legislation and institutional apparatus that is different from the one that the interviewers are familiar with?
Authors: Thank you for highlighting these points! Actually, the scoping review was applied for exploratory purposes. Consequently, barriers and enablers portrayed in this literature are brought up. We recognize that the use of scoping review for this purpose is a limitation of the study. Although the method served the research objectives as the first research step of the convergent parallel multimethod approach adopted, we now reinforce implications for future research in this regard, including the need for a systematic review of the literature, as discussed before in our comments to your last points. As stated by Munn et al. (2018): “A key difference between scoping reviews and systematic reviews is that in terms of a review question, a scoping review will have a broader “scope” than traditional systematic reviews with correspondingly more expansive inclusion criteria.” This is now made clearer in the first paragraph of Subsection 2.1 (part in blue). Additionally, we now explain better in different parts of the manuscript the connection of the three research steps conducted in the study, as explained before. We hope we could address better now your concern. The Scoping Review step did not concentrate on any geographic limitation or stages of development of countries or on the size of companies. Its focus was exclusively on what the literature presents for enablers and barriers, related to the central theme of the research. MSMEs were addressed in Steps 2 and 3. To address MSMEs, experts with backgrounds and experience in MSMEs, SSCM and I4.0 were accessed. In the introduction of the article, we indicate that this is a dilemma, because there is no understanding on the extant literature of how it affects the MSMEs supply chains, particularly in emerging countries. To lend the broader context, the panel of experts and the focus group discussions landed barriers and enablers to the reality of MSMEs. Although Brazil can be considered as representativeness in the group of emerging countries, indeed focus on one country is a limitation of our study. We now explain better this in different parts of the manuscript (highlighted in blue), particularly in Section 2 (Materials and Methods). Once again, thank you for shedding lights in this relevant issue!
Reviewer #1: The third aspect is the characterization of enablers and barriers. As I said, why much more enablers than barriers? What about the bias possible in the literature and the resulting scoping review. But the question here has much to do with the categories of the enablers and barriers. Are they symmetrical? On table 3, E3 and B10 and B11 seem to fit in a similar category.; the same between E31 and B4 or other ones.
In table 4 why are categories for barriers and enablers related? Why “social” for barriers and “people” for enablers. Some are the same like innovation, technology, legal, etc. But others are not so clear how they fit. For example, why institutional in the categories of enablers? Why a category of environment in barriers and one of sustainability in enablers? In table 6, the category Social as barrier and people in enablers are related or distinguishable?
Authors: Thank you for highlighting this important remark. Barriers and enablers are primarily originated as a result of the Scoping Review step. Results reveal 13 barriers and 31 enablers from the retrieved studies. In the panel of experts’ step, there was the possibility of listing new barriers or enablers that were not in the initial list, resulting in three extra barriers and another two enablers, thus totaling 16 barriers and 33 enablers. There is no symmetry between the barriers and enablers. This is now made clearer in the manuscript at the beginning of section 3 (Results) – second part of the first paragraph. We copy next this paragraph to ease your review, highlighting in blue the next text.
“Table 3 presents the 13 barriers (B) and 31 enablers (E) retrieved from the scoping review, added by the three extra barriers (EB) and two extra enablers (EE) obtained with the panel of experts. Appendix D presents the evaluation of the experts within the panel for each barrier and enabler. During this process, there was no purpose to balance the number of barriers and of enablers, or to develop a symmetry, but to retrieve the main ones available in the literature and refined with the aid of experts.”
We also thank you for bringing to us the perspective of symmetry between the categorization of barriers and enablers. Indeed, this makes sense and was a great suggestion! We reviewed our content analysis and verified that these categories can be symmetric, and this improves the presentation of the research findings. The “social” category was replaced by “people” and the “environment” category was replaced by “sustainability”, both in barriers, without loss of content. The “economic” and “stakeholders” categories are now analyzed from a single category entitled “OSCM related topics” for both barriers ad enablers. Tables 4 and 5 (previously labeled 5 and 6) contain these changes in blue. The previous table 4 was removed and replaced by the following text, as now we do have the symmetrical view for the barriers and enablers categories (in blue the new part).
“A complete analysis of all barriers and enablers for categorization was carried out through content analysis resulting in eight symmetrical categories: people, technology, innovation, institutional, OSCM related topics, legal, organization and sustainability.”
Reviewer #1: Some minor points or questions, now.
In the tables 7 and after, the interpretation for Di+Ri is rather straightforward. But it is not the case for Di-Ri.
Authors: Indeed, this is an important concern. We agree with your comment, and we have inserted improvements in the paper in this regard. We offer next a description of the main changed conducted.
1) At the end of subsection 2.3 (Focus Group), we describe the DEMATEL MAP to introduce the topic to the reader. It also embraces a new Figure 3 (The DEMATEL map). We copy below part of this content, highlighting in blue the new part to ease your review.
“ … Therefore, the Fuzzy-DEMATEL was applied in this research based on the adopted steps offered in ([42]; [45-48]) as follows:
- Create the correspondence of linguistic terms and values (see Table 2);
- Aggregate results and obtain a fuzzy pairwise direct-relation matrix (X);
- Normalize the direct-relation matrix and calculate the total relation matrix (T);
- Determine row and column sums from T;
- Determine the overall prominence and net effect values of factors (D+R and D-R);
- Formulate the DEMATEL cause-effect diagrams. Each step incorporates multiple mathematical evaluations. The prominence and net effect values of each factor are fuzzy-DEMATEL analysis outputs [47].
The final prominence value ranks the factors. Additional details on the fuzzy-DEMATEL methodology and the calculations appear in Table 2 and Appendix C.
From an ‘n × n’ identity matrix T, we have R being the sum of the rows and D being the sum of the columns of the matrix T, D+R is set to highlight, indicating the prominence of factors in the system, and D-R represents the importance for the influence of each factor (Figure 3).
-- Insert Figure 3 --
Figure 3. The DEMATEL MAP: adapted from [47].
2) We changed the heading of subsection 4.2 from “Prominence, influencers and resultant factors” to “Prominence (D+R), importance (D-R), and DEMATEL map discussion”
3) At the beginning of subsection 4.2 the following text is inserted, including table 14, as below. We copy below part of this content, highlighting in blue the new part to ease your review.
“The emerging key barriers and enablers are evaluated using the Fuzzy-DEMATEL method, and it helps to gain novel comprehensions with theoretical advances and practical implications. “D+R" score signifies the total cause and effect (prominence). The higher the value of the “D+R" the greater the influence of the barrier or enabler “y”. The comparative ranking of prominent factors based on the "D+R" scores appears in Table 13, separately for MSEs, MEs and for both combined. The value of “D-R" shows the net effect of the barrier or enabler “y”. The barrier or the enabler will be considered a cause (if positive) or a resulting (if negative) net effect (D-R). Consistent with [57], net effects can guide practical propositions as suggested in this paper. Companies would gain to focus on cause effects first and to attack resulting or consequent effects after [53, p.13]. The most influential (D-R positive) and consequent factors (D-R negative) are different according to size. This fact triggers separate propositions for practice and policymaking according to enterprise size. Barriers or enablers (factors) with a higher D-R value generate greater influence, being considered of higher priority or importance [47].
Table 13. D+R Results for Barriers and Enablers
|
Factor |
MSEs |
MEs |
MSMEs (Combined) |
|
Barrier |
B5>B1>B4 |
B5>B3>B8 |
B5>B3>B1 |
|
Enablers |
E4>E5>E6 |
E8>E1>E2 |
E8>E2>E1 |
Note: B1 – Lack of technical expertise; B3 - Resistance to change/change management practices and adopting innovation for society; B4 - Lack of investment in R&D; B5 - Cost of improvement & OSCM economic condition; B8 - Alternative resources and energy needs. E1 - Top management commitment + Strategic alignment; E2 - Employee’s empowerment + Knowledge sharing + Effective communication; E4 - Data-centered solutions + Consistent data flow; E5 - Interdisciplinary and holistic integration + Life cycle thinking and circular processes; E6 - Customer and supplier integration; E8 - Valuing R&D/Research Centers.
Once these cause factors (D-R values greater than 0) are calculated, table 14 ranks barriers and enablers for MSEs, MEs and MSMEs. Indications of cause and effect are shown in figures 8-13.”
Table 14. D-R Results for Barriers and Enablers (values > 0)
|
Factor |
MSEs |
MEs |
MSMEs (Combined) |
|
Barrier |
B7>B6>B8 |
B6>B7>B1 |
B7>B6>B1 |
|
Enablers |
E7>E1>E2 |
E1>E7 |
E7>E1>E2 |
Note: B1 – Lack of technical expertise; B6 - Lack of support from regulatory authority/poor legislation; B7 - Lack of commitment from top management; B8 - Alternative resources and energy needs. E1 - Top management commitment + Strategic alignment; E2 - Employee’s empowerment + Knowledge sharing + Effective communication; E7 - Governmental and institutional pressures; E8 - Valuing R&D/Research Centers.”
Reviewer #1: In several parts of the paper, the reference made to authors with numbers are used as subjects of the sentences, grammatically speaking. I find that odd. See line 35, 68, 69, 70, 74, 86, 90, 96, etc. I would expect the names of the author or first two or three.
Authors: We do appreciate your indication and your extra care in the notes within the text that we found attached to your comments. We made improvements based on all your notes, as mentioned next. We did struggle a bit with the format of the journal, which recommends that references must be numbered in order of appearance in the text, but now after your comment we did correct the inconsistencies.
We have also corrected the following issues pointed in your review:
by ... name of author/ or rephrasing – CORRECTED/DONE
Grammatically not correct, as all the sustition of the subject by a reference number – CORRECTED/DONE
I'm not used to this king of shorthand for referring to authors. See also line 74 for citation n. 15 – CORRECTED/DONE
point in excess?? – CORRECTED/DONE
i would suggest a comma between summarized and aided – CORRECTED/DONE
should be ":" and not "." after as follow – CORRECTED/DONE
Reviewer #1: The conclusion is good but could be improved in terms of interpretation of the results. What does it mean for industrial managers that focus on sustainability or even industry 4?
Authors: Once again, we thank you for your comments. We devoted in this revision a special attention to the presentation and interpretation of the results, with significant improvements in different parts of the manuscript. In section 2 (Materials and Methods) we improved the explanation of the three research steps, as well as the way they are interconnected, towards providing the reader a better understanding of how data was gathered, analyses and interpreted. In section 3 (Results) we clarified the categorization of barriers and enablers and included a discussion of the DEMATEL map for the barriers and enablers, segregating MSEs, MEs and MSMEs. We copy below this new content on this discussion to ease your review. At the end of section 3 (Results).
Based on Figure 3 (The DEMATEL map), barriers and enablers are distributed into four classifications: (i) key; (ii) minor key; (iii) indirect; (iv) independent. Table 12 presents the distribution for MSEs, MEs and for MSMEs, which is the central focus for analysis in this topic.
Table 12. The DEMATEL map discussion for Barriers and Enablers
|
Barriers |
MSEs |
MEs |
MSMEs (Combined) |
|
Key barriers |
B1, B6, B7, B8 |
B1, B6, B7 |
B1, B6, B7, B8 |
|
Minor key barriers |
none |
none |
none |
|
Indirect barriers |
B2, B3, B4, B5 |
B2, B3, B4, B5, B4, B8 |
B2, B3, B4, B5 |
|
Independent barriers |
none |
none |
none |
|
Enablers |
MSEs |
MEs |
MSMEs (Combined) |
|
Key enablers |
E1, E2 |
E1 |
E1, E2 |
|
Minor key enablers |
E7 |
E7 |
E7 |
|
Indirect enablers |
E3, E4, E5, E6, E8 |
E2, E8 |
E3, E4, E5, E6, E8 |
|
Independent enablers |
none |
E3, E4, E5, E6 |
none |
Note: B1 – Lack of technical expertise; B2 - Cybersecurity issues; B3 - Resistance to change/change management practices and adopting innovation for society; B4 - Lack of investment in R&D; B5 - Cost of improvement & OSCM economic condition; B6 - Lack of support from regulatory authority/poor legislation; B7 - Lack of commitment from top management; B8 - Alternative resources and energy needs. E1 - Top management commitment + Strategic alignment; E2 - Employee’s empowerment + Knowledge sharing + Effective communication; E3 - Internal innovation process; E4 - Data-centered solutions + Consistent data flow; E5 - Interdisciplinary and holistic integration + Life cycle thinking and circular processes; E6 - Customer and supplier integration; E7 - Governmental and institutional pressures; E8 - Valuing R&D/Research Centers.
The classification of barriers does not show much variation between MSEs, MEs, and combined MSMEs. B8 appears as an indirect barrier for MEs and as a key barrier both for MSEs and in combined MSMEs, where in addition to B8, barriers B1, B6 and B7 also appear as key barriers for MSMEs. B8 is classified as an indirect barrier in MEs. It is noteworthy that there is no classification of barriers as minor key barriers or independent barriers.
Regarding enablers, there is full alignment between the results for MSEs and combined MSMEs. E1 is a key enabler. E2 appears as key enabler for MSEs and combined MSMEs, and as indirect enabler for MEs. E3, E4, E5 and E6 appear as independent enablers for MEs. E7 always appears as a minor key enabler. There are not enablers classified as independent for MSEs and MSMEs.”
In subsection 4.2 (Prominence (D+R), importance (D-R), and DEMATEL map discussion) we reinforced the discussion on D-R, as requested before, with new contents and a new table 14. Moreover, we also enriched P14 and P15 to address your comment. We copy below part of this content, highlighting in blue the new part to ease your review.
“P14 – MSMEs priority actions to circumvent cause barriers should gear at mitigating the lack of support from regulatory authority/poor legislation (B6), the lack of commitment from top management (B7), and the lack of technical expertise (B1). The consequent barriers should be tackled later. B1, B6 and B7 are key barriers and should receive priority in the treatment by the company and by other stakeholders involved, such as governments and educational institutions and technical training. The Alternative resources and energy needs (B8) barrier also draws attention as key barriers for MSEs and as an indirect barrier for MEs. In the combined MSMEs predominates as a key barrier, which allows us to attest greater difficulty for MSEs to overcome this barrier due to less formal access to clean energy financing programs, especially in developing countries. One possible suggestion is to reduce real financial guarantees so that MSEs can access, in scale, the financing available or that may become available.”
P15 – Policy and action programs for MEs should be directed in priority to enhance the cause enablers of top management commitment and strategic alignment (E1), and to burst governmental and institutional support (E7), before addressing consequential enablers. The enabler Top management commitment + Strategic alignment (E1) is considered a key enabler. Another highlight is Employee's empowerment + Knowledge sharing + Effective communication (E2) considered indirect enabler for MEs and key enabler for MSEs and combined MSMEs. This result makes a lot of sense when looking at key barriers because these two enablers are directly associated mainly with key barriers B1 and B7.”
Finally, in the conclusion section, we inserted a new paragraph with additional practical opportunities for manager´s reflection, as follows:
“The results, discussions, and propositions of this research offer additional practical opportunities for managers' reflection. It is challenging to look at the implementation of I4.0 technologies with a sustainable perspective on the supply chain. The interaction with the environment outside the industry is an additional challenge. In the proposals, there are examples that can directly help managers, such as: (i) implementation of policies; (ii) establishment of action programs; (iii) identification of key factors, prioritizing the treatment of barriers and enablers; (iv) offer of concrete arguments for dealing with other interest groups involved such as governments, as in the case of public policies to encourage the use of clean energy sources”.

Reviewer 2 Report
The paper deal with an interesting topic related to Industry 4.0, supply chain and MSMEs.
The topic is original and it's proposed timely.
The structure of the paper is clear. Motivations and novelty are proposted in the Introduction.
The methodology is robust. Results and discussions are appropriate and are linked to the aim and RQs.
Conclusions are consistent.
Author Response
REVIEWER#2: COMMENTS & SUGGESTIONS FOR AUTHORS
Reviewer #2: The paper deal with an interesting topic related to Industry 4.0, supply chain and MSMEs. The topic is original and it's proposed timely.
Authors: Thank you for acknowledging that the topic is interesting, original and it's proposed timely. Indeed, this motivates us even more to continue our research.
Reviewer #2: The structure of the paper is clear. Motivations and novelty are proposted in the Introduction. The methodology is robust. Results and discussions are appropriate and are linked to the aim and RQs. Conclusions are consistent.
Authors: Thank you very much for the comments and compliments on our paper! We take advantage of your encouraging words that also encouraged us to introduce improvements to carry out a qualitative upgrade in the article. We understand that the comments of three reviewers stimulated our careful reflection, making it possible to leverage positive points and make improvements in topics that presented this opportunity. We hope you will enjoy the new version of the manuscript.

Reviewer 3 Report
The authors chose a very interesting and topical theme for the paper, have relevant data and used appropriate research methods. Nevertheless, the text in its current form is not good and is very demanding to the reader, who must be very detailed and comprehensively know most of the theoretical starting points for the text in order to understand it. I think the authors are dealing with an overly extensive problem and such research would be more appropriate for preparing a book than an article. The choice of too broad a problem is also the reason for most of the problems of the current version of the text.
I think it makes sense to give authors the opportunity to correct the text in a more readable form. Of course, this means that authors have to do a strong revision of the text. It would probably make sense for authors to make two or more texts from the current text - so they would have more room to supplement the content work and a more specific definition of the results.
Author Response
REVIEWER#3: COMMENTS & SUGGESTIONS FOR AUTHORS
Reviewer #3: The authors chose a very interesting and topical theme for the paper, have relevant data and used appropriate research methods.
Authors: We appreciate very much your nice comment about our paper. This stimulates us to continue to work hard in our research. Thank you!
Reviewer #3: Nevertheless, the text in its current form is not good and is very demanding to the reader, who must be very detailed and comprehensively know most of the theoretical starting points for the text in order to understand it. I think the authors are dealing with an overly extensive problem and such research would be more appropriate for preparing a book than an article. The choice of too broad a problem is also the reason for most of the problems of the current version of the text. I think it makes sense to give authors the opportunity to correct the text in a more readable form. Of course, this means that authors have to do a strong revision of the text. It would probably make sense for authors to make two or more texts from the current text - so they would have more room to supplement the content work and a more specific definition of the results.
Authors: We thank you for your detailed and rich comments. We made several changes to the manuscript to address all of them. Not only did we rephrased different parts to improve the reader's understanding but added additional content and analyses to enrich the theoretical and practical contributions of the research. The main changes are highlighted in blue in the new version of the manuscript and described below to ease the reviewing process.
We refined different theoretical starting points to help readers in understanding the research proposal, methods, and research findings´ discussion. In this regard, new references were included, as follows:
New references:
- Saroha, M., Garg, D., Luthra, S. (2021). Identification and analysis of circular supply chain management practices for sustainability: a fuzzy-DEMATEL approach. International Journal of Productivity and Performance Management. Vol ahead-of-print. ahead-of-print. DOI: 10.1108/IJPPM-11-2020-0613
- Munn, Z., Peters, M.D.J., Stern, C. et al. Systematic review or scoping review? Guidance for authors when choosing between a systematic or scoping review approach. BMC Medical Research Methodology, 18(1), 143. DOI: 10.1186/s12874-018-0611-x
- Flynn, B.B., Sakakibara, S., Schroeder, R.G., Bates, K.A., Flynn, E.J. (1990). Empirical research methods in operations management. Journal of operations management, 9(2), 250-284. DOI: 10.1016/0272-6963(90)90098-X
- Han, Y., Deng, Y. (2018). An enhanced fuzzy evidential DEMATEL method with its application to identify critical success factors. Soft Computing 22(15), 5073–5090. DOI: 10.1007/s00500-018-3311-x
- Zadeh, L.A. (1965). Fuzzy sets. Information and Control, 8(3), 338-353. DOI: 10.1016/S0019-9958(65)90241-X
- Creswell, J.W., Creswell, J.D. (2018). Research Design: Qualitative, Quantitative, and Mixed Methods Approaches. Los Angeles, CA: Sage.
The revision of the text included more details on the conduction of the multimethod approach. We dedicated more room to the description of each of the three research steps and their relation to the generation of the research findings. The rationale to transition from one step to the next one is also better explained. Now we emphasize in step 1 that its aim was a broad view on barriers and enablers to integrate I4.0 and sustainability in supply chains, and that the next two research steps were specific to MSMEs, narrowing the main scope of the paper. Research limitations are now better explained and highlighted, which are taken up again at the conclusion section of the paper reinforcing recommendations for future research. The main changes in these regards are highlighted in sections 1, 2, and 5.
Additionally, we included more details in our research findings and additional discussions and interpretations to improve the contributions of the paper. We defined better the categorization of barriers and enablers and improved the interpretation for Di-Ri. A DEMATEL map discussion is now available, as well as a more complete P14 and P15. The main changes in these regards are highlighted in sections 3, 4, and 5.
We hope that the strong revision done in the previous version of the manuscript could address all your pertinent concerns. Now, we believe that the new version of the manuscript proves readers with a more direct and complete text, reinforcing the research´s scope, method, findings, and contributions. Changes also were conducted to ease the readability and understanding of the paper towards providing a more readable form.

Round 2
Reviewer 3 Report
The authors took into account most of the comments from the first phase of the review process. The text could be further improved, but a new version of the text is already suitable for publication.